# Pyruvate kinase type M2 promotes tumour cell exosome release via phosphorylating synaptosome-associated protein 23

Yao Wei[1,2,*], Dong Wang[1,*], Fangfang Jin[1,*], Zhen Bian[3,*], Limin Li[1], Hongwei Liang[1], Mingzhen Li[1], Lei Shi[3], Chaoyun Pan[1], Dihan Zhu[1], Xi Chen[1], Gang Hu[2], Yuan Liu[3], Chen-Yu Zhang[1] & Ke Zen[1,3]

Tumour cells secrete exosomes that are involved in the remodelling of the tumour–stromal environment and promoting malignancy. The mechanisms governing tumour exosome release, however, remain incompletely understood. Here we show that tumour cell exosomes secretion is controlled by pyruvate kinase type M2 (PKM2), which is upregulated and phosphorylated in tumours. During exosome secretion, phosphorylated PKM2 serves as a protein kinase to phosphorylate synaptosome-associated protein 23 (SNAP-23), which in turn enables the formation of the SNARE complex to allow exosomes release. Direct phosphorylation assay and mass spectrometry confirm that PKM2 phosphorylates SNAP-23 at Ser95. Ectopic expression of non-phosphorylated SNAP-23 mutant (Ser95→Ala95) significantly reduces PKM2-mediated exosomes release whereas expression of selective phosphomimetic SNAP-23 mutants (Ser95→Glu95 but not Ser20→Glu20) rescues the impaired exosomes release induced by PKM2 knockdown. Our findings reveal a non-metabolic function of PKM2, an enzyme associated with tumour cell reliance on aerobic glycolysis, in promoting tumour cell exosome release.

[1] State Key Laboratory of Pharmaceutical Biotechnology, Jiangsu Engineering Research Center for MicroRNA Biology and Biotechnology, Nanjing Advanced Institute for Life Sciences, School of Life Sciences, Nanjing University, Nanjing, Jiangsu 210093, China. [2] School of Medicine and Life Sciences, Nanjing University of Chinese Medicine, Nanjing, Jiangsu 210023, China. [3] Center for Immunology, Inflammation and Infectious Diseases & Department of Biology, Georgia State University, Atlanta, Georgia 30302, USA. * These authors contributed equally to this work. Correspondence and requests for materials should be addressed to G.H. (email: ghu@njutcm.edu.cn) or to Y.L. (email: yliu@gsu.edu) or to C.-Y.Z. (email: cyzhang@nju.edu.cn) or to K.Z. (email: kzen@nju.edu.cn).

As a mechanism to communicate with the microenvironment, tumour cells actively release large quantity of extracellular vesicles (EVs), including exosomes, microvesicles (MVs) or microparticles, and apoptotic bodies. These tumour-released EVs, which are abundant in the body fluids of patients with cancer, play a critical role in promoting tumour growth and progression[1,2]. For example, NCI-H460 tumour cells actively release MVs containing EMMPRIN, a transmembrane glycoprotein highly expressed by tumour cells, MV-encapsulated EMMPRIN that facilitates tumour invasion and metastasis via stimulating matrix metalloproteinase expression in fibroblasts[3]. Tumour cell exosomes also deliver active Wnt proteins to regulate target cell β-catenin-dependent gene expression[4]. Cancer cell-derived microparticles bearing P-selectin glycoprotein ligand 1 accelerate thrombus formation in vivo, and by targeting P-selectin glycoprotein ligand 1 researchers were able to prevent thrombosis[5]. While these studies are exciting and suggest potential strategies for blocking metastasis, the mechanism underlying the active exocytosis of exosomes by tumour cells, however, remains unclear. Previous studies suggest that cellular exosome secretion activity is increased during tumorigenesis[6,7], but the molecular basis for switching on the exocytosis process in tumour cells requires further clarification.

The mechanisms that govern cell endosomal secretion have been extensively studied. Exosomes share structural and biochemical characteristic with intraluminal vesicles of multivesicular endosomes (MVEs). Studying trafficking of proteolipid protein in Oli-neu cells, Trajkovic et al.[8] reported that the sphingolipid ceramide played a key role in triggering budding of exosomes into MVEs, and the release of exosomes was reduced after the inhibition of ceramide synthesis. Furthermore, Kosaka et al.[9] found that neutral sphingomyelinase 2 was directly involved in promoting tumour cell endosomal secretion. Using an RNAi screen, Ostrowski et al.[10] identified the role of Rab GTPases in promoting exosome secretion: among the small GTPases, Rab27a and Rab27b were involved in MVE docking to the plasma membrane. Like other cells, tumour cells employ the soluble N-ethylmaleimide-sensitive fusion factor attachment protein receptor (SNARE) complex that many cell types utilize in the exocytic release of exosomes[11]. The SNARE complex is comprised of proteins on membrane of budding vesicles (v-SNAREs) and proteins on the cell's membrane (t-SNAREs). The v-SNAREs and t-SNAREs enable the apposition of the vesicle and cell membranes and the subsequent fusion of the two membranes thereby mediating vesicle exocytosis. In tumour cells, the SNARE complex includes syntaxin-4 (ref. 12) and SNAP-23 (ref. 13) serving as t-SNAREs, while VAMP-2 (ref. 14) and VAMP-8 (refs 12,15) represent candidates for v-SNAREs. Phosphorylation of SNAP-23 not only directly increases cell exocytosis[16,17] but also promotes association with other SNARE proteins, thereby allowing the formation of the stable SNARE complex to enhance cell exocytosis[18]. In mast cells, SNAP-23 has been reported to be phosphorylated by IκB kinase (IKK) to promote exocytosis[19,20]. However, the kinase that phosphorylates SNAP-23 in the tumour cell has not been identified.

In the present study, we demonstrate that PKM2, an enzyme involved in the tumour cell's reliance on aerobic glycolysis (Warburg effect), plays a critical role in promoting the release of exosomes from the tumour cell. Specifically, we identify SNAP-23, which controls the docking and release of secretory granules or exosome-containing multivesicular bodies, is a substrate of PKM2 in tumour cells. During exosome secretion, phosphorylated PKM2 forms a dimer structure with low pyruvate kinase activity but high protein kinase activity[21] and then associates with SNAP-23 near cell's membranes, leading to SNAP-23 phosphorylation at Ser95 and upregulation of tumour

cell exosome release. We conclude that PKM2, following phosphorylation and dimerization, plays an essential role in not only switching tumour cell metabolism from oxidative phosphorylation to aerobic glycolysis, but also promoting tumour cell exosome secretion via directly phosphorylating SNAP-23.

## Results

**Exosome secretion requires high level of aerobic glycolysis.** Exosomes released by various cell types were isolated from cell culture medium using exosome isolation kit and analysed by transmission electron microscopy and western blot (WB) using antibodies against exosomal marker proteins. As shown in Fig. 1a, tumour cells actively release exosomes, a double membrane vesicle with 50–100 nm size, into the culture medium. The immune-gold label showed that exosomes expressed membrane protein CD63 (Fig. 1a, right lower panel). Expression of additional marker proteins such as CD63, Tsg101 and CD9 was validated by WB analysis (Fig. 1b). To monitor the concentration of exosomes released by tumour cells and non-tumour primary culture cells, a Nanosight NS 300 system (NanoSight) was used to track the release of exosomes (Fig. 1c). Nanoparticle tracking analysis (NTA) confirmed that the sizes of released exosomes are around 100 nm. In agreement with previous reports, we found that tumour cells (SW480, Hela, A549 and HepG2 cells) generally displayed more active exosome secretion than non-tumour mammalian cells, mouse primary myoblast and mammary epithelial cell (MEC) (Fig. 1d). Interestingly, the increased exosome release by tumour cells is positively correlated to the higher aerobic glycolysis (Fig. 1e). In line with the positive correlation between aerobic glycolysis and exosome secretion observed in Fig. 1f, we found that glycolysis level was positively correlated with the amount of exosome release in tumour cells (Supplementary Fig. 1). To further examine the potential link between the exosome secretion and the aerobic glycolysis flow in the tumour cells, we treated A549 cells and HepG2 cells with aerobic glycolysis inhibitor shikonin[22] or aerobic glycolysis promoter tumour necrosis factor α (ref. 23), and then determined the alteration of exosome release from tumour cells. The results clearly showed that suppressed aerobic glycolysis by shikonin inhibited, whereas activated aerobic glycolysis by TNFα enhanced, the release of exosomes by tumour cells (Fig. 1g). Identification of the dependence of exosome secretion process on cell glycolysis is in agreement with previous findings in other cell types[22,24].

Previous reports showed that epidermal growth factor (EGF) and oleanolic acid (OA) can enhance or inhibit exosome secretion process[25,26], respectively. We next treated the A549 tumour cells with EGF or OA and examined whether the effect of these reagents was mediated through alteration of tumour cell aerobic glycolysis. As shown in Fig. 1h, EGF and OA significantly enhanced and suppressed aerobic glycolysis in A549 cells, respectively. We next examined whether the effect of EGF and OA was mediated through alteration of tumour cell aerobic glycolysis. As shown in Fig. 1i, EGF and OA significantly enhanced and suppressed aerobic glycolysis in A549 and HepG2 cells, respectively. Moreover, the effect EGF and OA on promoting or inhibiting A549 and HepG2 cells exosome exocytosis was abolished by decreasing or increasing aerobic glycolysis, respectively. These results collectively suggest that release of exosomes in tumour cells is dependent on cellular aerobic glycolysis.

**PKM2 plays a critical role in tumour cell exocytosis.** PKM2 expression has been widely regarded as an important molecular

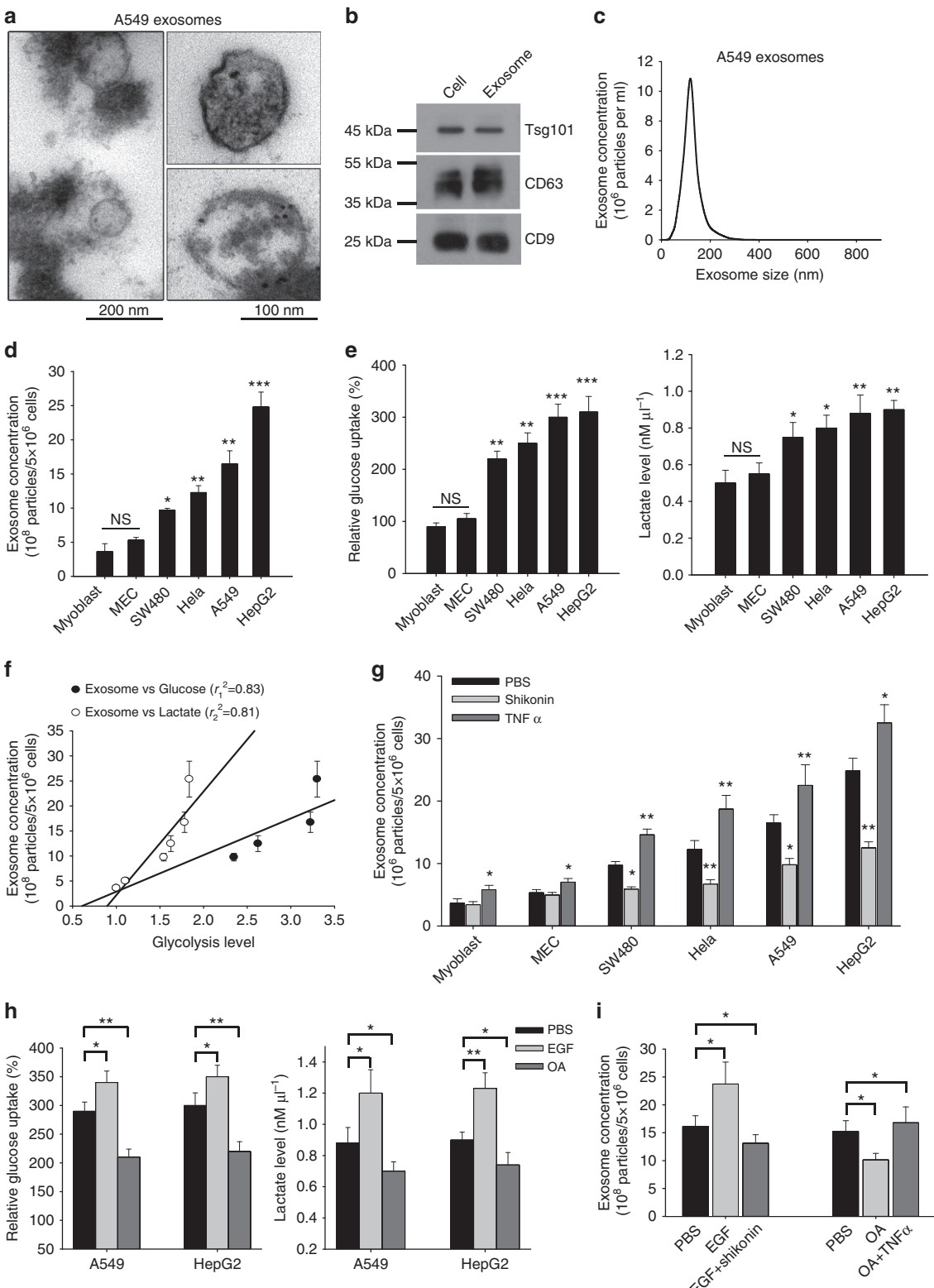

**Figure 1 | Release of exosomes by tumour cells depends on aerobic glycolysis.** (**a–c**) Isolated exosomes from A549 cells assessed by transmission electron microscopy (**a**), WB (**b**) and NTA (**c**). The lower right image in **a** represents the immune-gold labelling of CD63 in an exosome. (**d**) Tumour cells release more exosomes than non-tumour cells. (**e**) Positive correlation between exosome secretion and aerobic glycolysis. (**f**) Linear regression between glucose uptake ($r_1^2 = 0.83$) and lactate level ($r_2^2 = 0.81$). (**g**) Release of exosomes is dependent on cellular aerobic glycolysis. Cells were treated with shikonin ($1\,\mu M$) or tumour necrosis factor $\alpha$ ($5\,ng\,ml^{-1}$) to inhibit or promote aerobic glycolysis. Exosome concentration was measured $24\,h$ post-treatment. (**h**) Effect of EGF and OA on cell aerobic glycolysis. Note that EGF ($10\,ng\,ml^{-1}$), an enhancer of exosome release, increases aerobic glycolysis, while OA ($10\,\mu g\,ml^{-1}$), an inhibitor of exosome release, decreases aerobic glycolysis. (**i**) EGF and OA regulate A549 cell exosome release via altering cellular aerobic glycolysis. Data are presented as the mean ± s.e.m. and represent at least three independent experiments with three replicates per data point. NS, no significance. $*P<0.05$, $**P<0.01$, $**P<0.001$ as determined by the one-way ANOVA test.

feature of tumour development[27]. In tumour cells, PKM2 forms a dimer, which is catalytically inactive for conversion of phosphoenolpyruvate (PEP) to pyruvate and production of ATP[28,29]. Lowering pyruvate formation provides a growth advantage for tumour progression as blocking production of pyruvate helps to channel the glycolytic intermediates to biosynthesis to meet the demands for tumour cell proliferation. Given that PKM2 plays a key role in switching tumour cell metabolic status from oxidative phosphorylation to aerobic glycolysis and PKM2 level is increased during tumorigenesis[30,31], we next tested whether tumour cell exosome release is correlated with the PKM2 expression. In agreement with previous reports, tumour cells contained significantly higher PKM2 level than non-tumour cells (Fig. 2a). In line with the correlation between aerobic glycolysis and exocytosis observed in Fig. 1, we found that level of PKM2 was positively correlated with the amount of

exosome release in tumour cells (Fig. 2b). In a similar manner, tumour cells also showed significantly higher phosphorylated PKM2 (p-PKM2) level than non-tumour cells (Fig. 2c), and the p-PKM2 level was positively correlated with the amount of exosome release in tumour cells (Fig. 2d). Interestingly, treating A549 lung carcinoma cells with OA (Fig. 2e) or EGF (Fig. 2f), which enhanced or suppressed tumour cell exosome release, respectively, we found that the level of PKM2 in A549 cells was dose-dependently decreased by OA or increased by EGF.

To determine whether PKM2 particularly p-PKM2 plays a role in modulating the release of exosomes from tumour cells, we assessed exosome release after knocking down PKM2 level in A549 and Hela tumour cells via PKM2 siRNA or overexpressing PKM2 in myoblasts and MEC via HA-tagged PKM2-expressing plasmid. As shown in Supplementary Fig. 2, knockdown or overexpression of cellular PKM2 levels decreased or enhanced

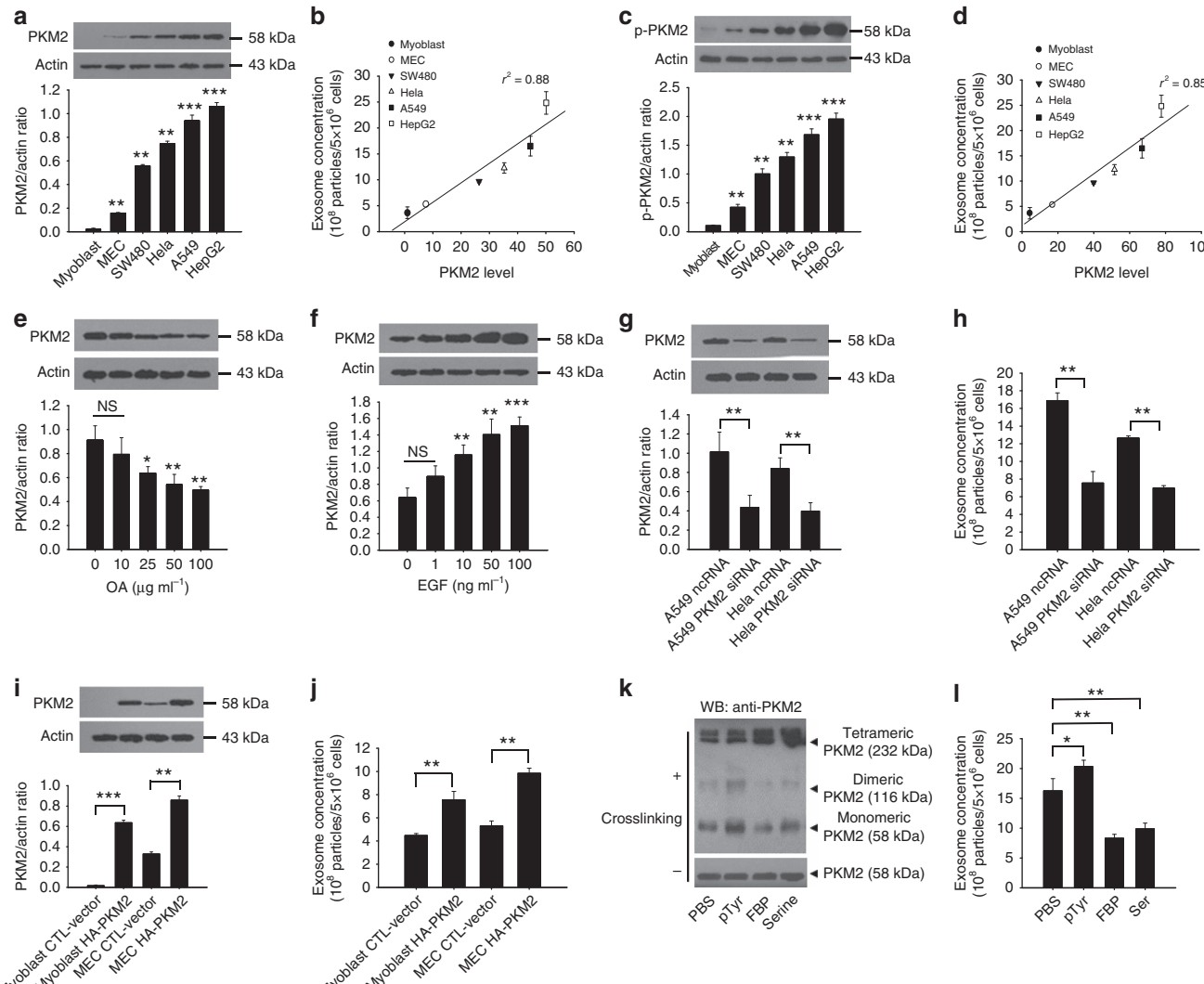

**Figure 2 | PKM2 plays a critical role in release of exosomes in tumour cells.** (**a**) Relative level of PKM2 in tumour or non-tumour cells. (**b**) Linear regression represents a positive correlation between PKM2 levels in different cell lines with exosome secretion. (**c**) Phosphorylated PKM2 level in tumour or non-tumour cells. (**d**) Linear regression represents a positive correlation of phosphorylated PKM2 level in different cell lines with exosome secretion. (**e**) OA, an inhibitor of exosome release, decreases PKM2 level. (**f**) EGF, an enhancer of exosome release, increases PKM2 level. (**g,h**) Knockdown of PKM2 in A549 and HeLa tumour cells via PKM2 siRNA (**g**) reduces the release of exosomes (**h**). (**i,j**) Overexpression of PKM2 in mouse primary myoblast cells and mammary epithelial cells (MEC) via transfection with HA-PKM2-expressing plasmid (**i**) increases the release of exosomes (**j**). (**k**) Effect of pTyr, FBP and serine on the switch of PKM2 from tetrameric formation to dimeric formation in A549 tumour cells. (**l**) Effects of pTyr (100 μM), FBP (500 μM) and serine (5 mM) on exosome release in A549 tumour cells. Data are presented as the mean ± s.e.m. of three independent experiments. *$P < 0.05$. **$P < 0.01$. ***$P < 0.001$ as determined by the one-way ANOVA test (two-tailed $t$-test for **g**–**j**).

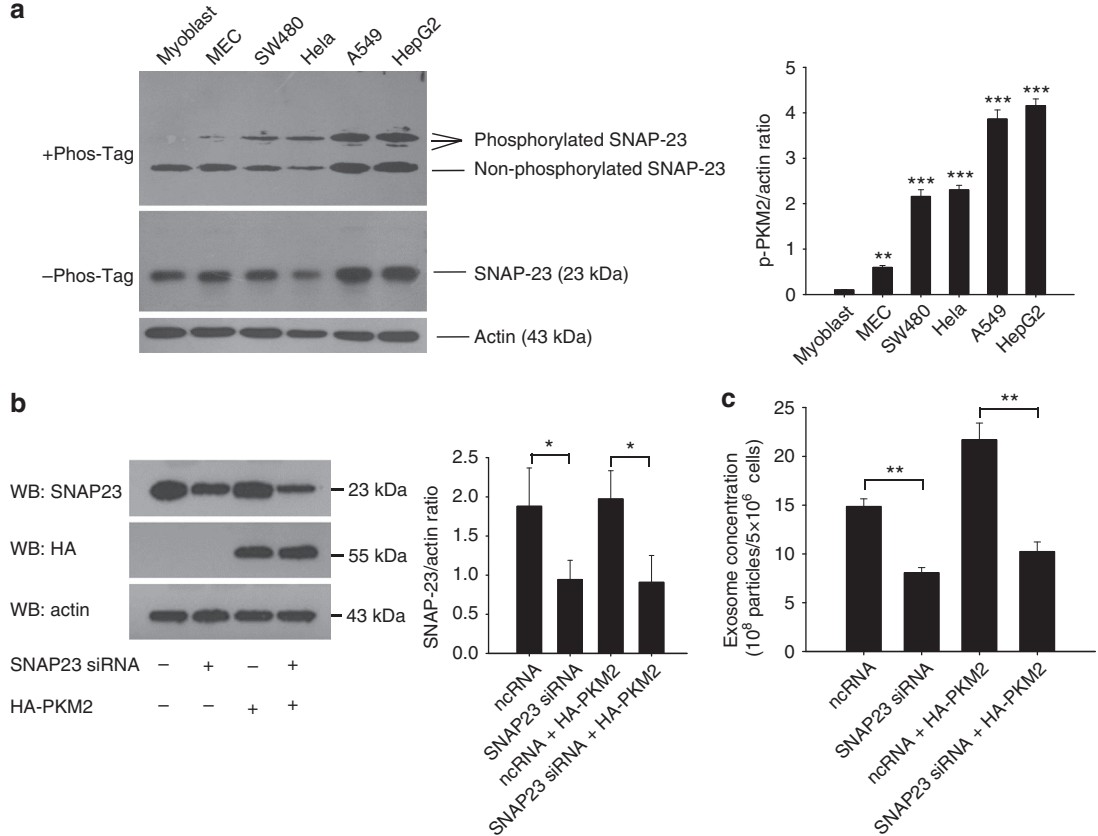

**Figure 3 | PKM2-promoted exosome release in tumour cells is dependent on SNAP-23.** (**a**) Relative level of SNAP-23 and phosphorylated SNAP-23 in tumour or non-tumour cells. (**b**) Knockdown of SNAP-23 and overexpression of PKM2 in A549 tumour cells via SNAP-23 siRNA and HA-PKM2-expressing plasmid, respectively. (**c**) Knockdown of SNAP-23 decreases PKM2-promoted release of exosomes in A549 cells. Data are presented as the mean ± s.e.m. of three independent experiments. *$P < 0.05$. **$P < 0.01$. ***$P < 0.001$ as determined by the one-way ANOVA test.

p-PKM2 levels. NTA results clearly showed that knockdown of PKM2 in A549 and Hela cells (Fig. 2g) strongly reduced the release of exosomes (Fig. 2h). In contrast, overexpression of HA-tagged PKM2 (HA-PKM2) in non-tumour cells (Fig. 2i) markedly enhanced the release of exosomes (Fig. 2j). Given that apoptotic cells secrete more exosomes than healthy cells and aerobic glycolysis inhibitor or PKM2 knockdown may affect cell apoptosis, we next determined the effects of shikonin and PKM2 knockdown on cell apoptosis using flow cytometry. As shown in Supplementary Fig. 3, both shikonin treatment and PKM2 knockdown significantly increased early or late apoptosis of A549 cells. Considering that total exosome release from A549 cells is decreased after shikonin treatment or PKM2 knockdown, increase of cell apoptosis by shikonin treatment or PKM2 knockdown further demonstrates that PKM2-mediated aerobic glycolysis promotes tumour cell exosome release. Furthermore, given that switching the behaviour of PKM2 from a tetramer form to a dimer form increases the initial steps of tumour cell aerobic glycolysis and promotes tumour progression[21,32], we treated A549 cells with pTyr, a phosphotyrosine peptide that can promote PKM2 dimeric formation[33], or fructose 1,6-bisphosphate (FBP) and serine, two molecules that enhance PKM2 tetrameric formation[34]. To assess the dimeric or tetrameric formation of PKM2, chemical crosslinking reaction was carried out to maintain the polymer structure before WB analysis[21]. Parallel samples without crosslinking treatment were included as loading controls. As expected, pTyr treatment increased the level of PKM2 dimer (116 kDa), while FBP and serine enhanced tetrameric formation (232 kDa) in A549 cells

(Fig. 2k). Consistent with the configuration of PKM2 either facilitating or reducing exosome exocytosis, pTyr, induced dimeric PKM2, increased tumour cell exosome exocytosis, while FBP and serine, which induced tetrameric PKM2, significantly decreased tumour cell exosome exocytosis (Fig. 2l).

In addition, through assaying the level change and the effect on secretion exosomes of PKM1, we found that pyruvate kinase activity of PKM might be not relevant to tumour cell exosome secretion. As shown in Supplementary Fig. 4, overexpression or knockdown of PKM1 in Hela and A549 cells displayed no effect on the release of exosomes from tumour cells. Taken together, these results strongly argue that PKM2, particularly phosphorylated PKM2 which easily dimerizes, plays an essential role in promoting the release of exosomes.

**PKM2-promoted exosome release is dependent on SNAP-23.** As a critical component of general cell exocytosis machinery, SNAP-23 has been widely reported to be involved in controlling cell exocytosis[13,16,17]. We isolated exosomes released by SW480, A549, Hela, 293 T and LLC cells, and then performed mass spectra analysis and an isobaric tags for relative and absolute quantitation (iTRAQ) assay for protein expression profiling. As expected, protein profiling analysis showed that SNAP-23 was the only component of SNARE complex detected in exosomes derived from all the five cell lines (Supplementary Table 1). To test whether PKM2-promoted tumour cell exocytosis is through SNAP-23-mediated exocytic machinery, we first compared the levels of SNAP-23 and phosphorylated SNAP-23 (p-SNAP-23) in tumour or non-tumour cells (Fig. 3a). As shown, tumour cells

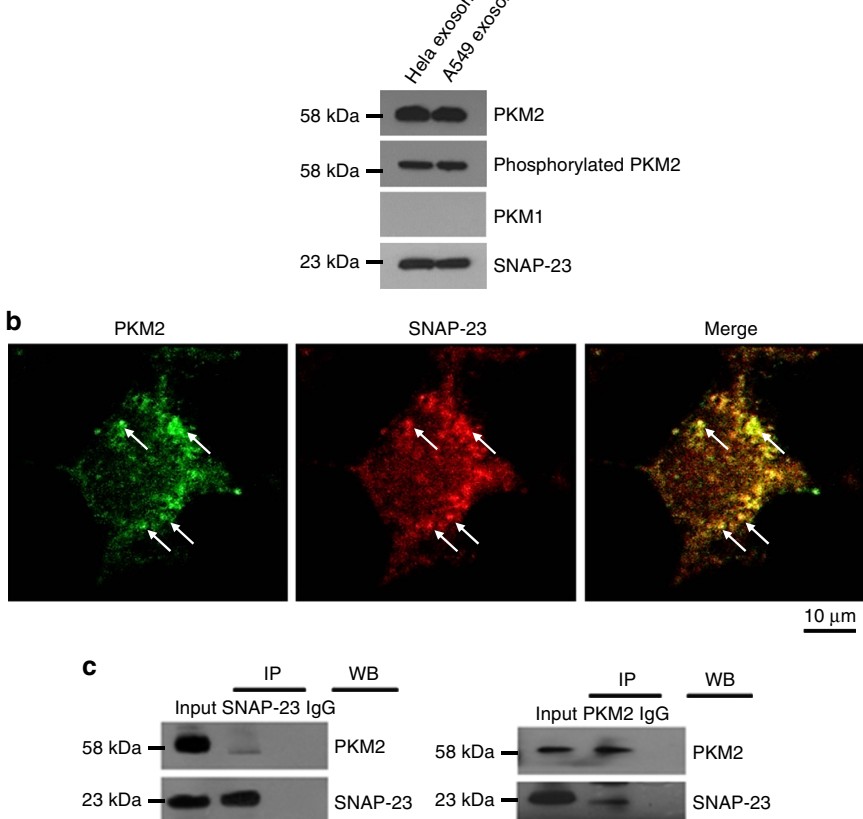

**Figure 4 | Association of PKM2 with SNAP-23 in tumour cell exosomes.** (**a**) WB analysis of isolated exosome fractions from Hela cells and A549 cells for PKM1, PKM2, p-PKM2 and SNAP-23 expression, 50 μg proteins of each sample were loaded. (**b**) Double staining of A549 cells with anti-PKM2 (green) and anti-SNAP-23 (red) antibodies. Arrows indicated the small punctate structures where SNAP-23 and PKM2 were co-localized. (**c**) Cross-IP and cross-blot of PKM2 and SNAP-23 in isolated exosome fraction from A549 cells using anti-SNAP-23 and anti-PKM2 antibodies, respectively. Immunoprecipitation using normal IgG served as controls.

particularly the cells with high capacity to secrete exosomes, had a significantly higher level of p-SNAP23 compared with myoblast and MEC. Next, we knocked down SNAP-23 in A549 cells and then assessed the release of exosomes. In a separate experiment, A549 cells were also transfected with HA-PKM2-expressing plasmid or co-transfected with HA-PKM2-expressing plasmid plus SNAP-23 siRNA. As shown in Fig. 3, knockdown of SNAP-23 (Fig. 3b) not only significantly blocked the release of exosome in A549 cells but also almost completely abolished the promotion of exosome release by PKM2 overexpression (Fig. 3c). The results suggest that PKM2-promoted exosome release in tumour cells is through SNAP-23-mediated mechanism.

Similar results were also observed when exosomes were isolated using sequential centrifugation method[35]. As shown in Supplementary Fig. 5, exosome release from A549 cells and Hela cells was significantly decreased after knocking down PKM2 via PKM2 siRNA (Supplementary Fig. 5a). In contrast, after overexpression of PKM2 via transfection with HA-PKM2-expressing plasmid, exosome secretion from myoblast and MEC was markedly enhanced (Supplementary Fig. 5b).

**PKM2 is associated with SNAP-23 during exocytosis process.** Further analysis of the protein profiling results showed that SNAP-23, VAMP3 and VAMP7, components of SNARE complex, as well as the exocytosis-related Rab small GTPases such as Rab1A and Rab2A, are present in the exosome fraction secreted by SW480, A549, Hela, 293 T and LLC cells (Supplementary Table 1). Based on the model of formation of endosomes,

exosomes and MVEs[36–38], the membranes of these vesicular structures share many protein and lipid components. However, to our surprise, we also found a considerable level of PKM2, but no PKM1, in the exosome fraction. Further analysis of isolated exosomes from tumour or non-tumour cells, we found that phosphorylated PKM2 was also associated with exosomes secreted by tumour cells (Supplementary Fig. 6). Identification of PKM2 in secreted exosomes has been also reported by Buschow *et al.*[39], whose results showed that PKM2 was co-immunoprecipitated with MHC II molecules in detergent-solubilized exosomes. The exosome protein profiling data support our hypothesis that PKM2 but not PKM1 is involved in modulating tumour cell exosome release.

We next analysed the expression and localization of PKM2 and SNAP-23 in A549 cells during exosome release using WB and immunofluorescence (IF) staining assays. As shown in Fig. 4a, WB analysis of exosome fraction isolated from culture medium of A549 cells and Hela cells detected SNAP-23 and PKM2, particularly phosphorylated PKM2 (p-PKM2), but not PKM1, confirming the protein profiling data that secreted exosomes contained SANP-23, PKM2 and p-PKM2 but not PKM1. Double IF staining of SNAP-23 and PKM2 in A549 cells further showed a considerable co-localization of SNAP-23 (red) and PKM2 (green) at certain punctate structures close to or on the cell surface (Fig. 4b, arrows), suggesting that SNAP-23 and PKM2 may be associated with each other when secretory granules or exosome-containing multivesicular bodies dock at the cell's plasma membranes.

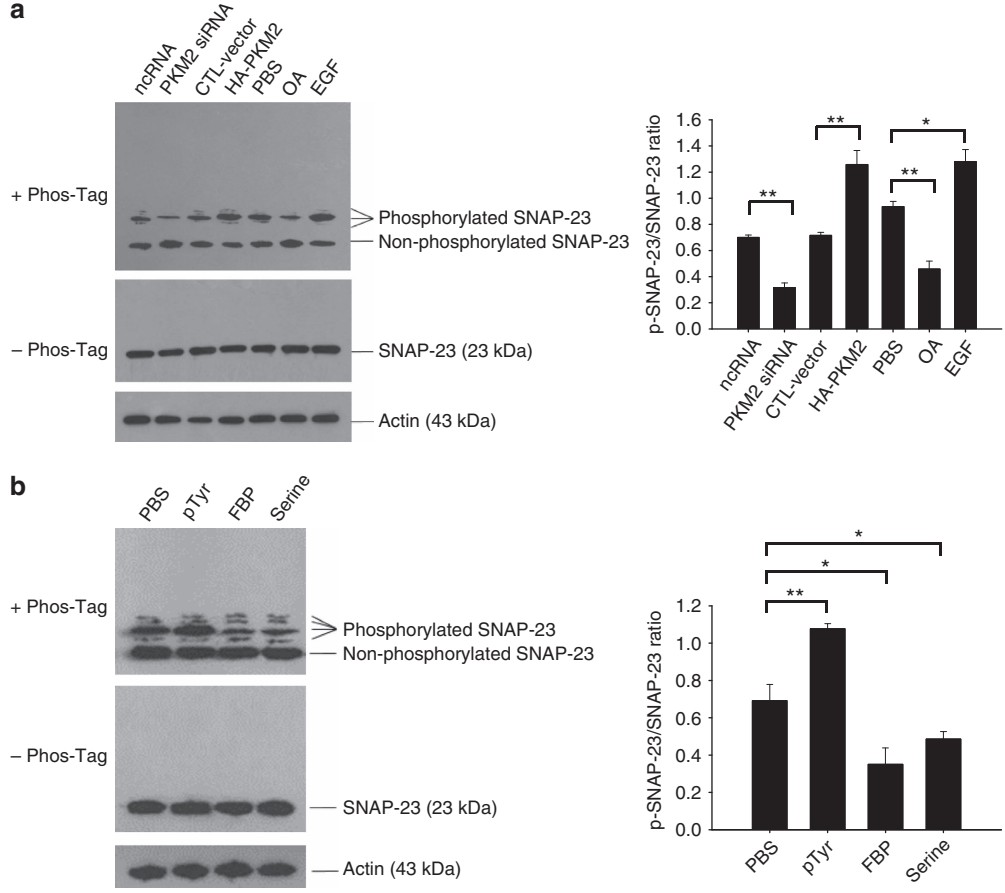

**Figure 5 | PKM2 promotes the release of exosomes in tumour cells via phosphorylating SNAP-23.** The levels of phosphorylated or non-phosphorylated SNAP-23 in A549 cells assessed by SDS-PAGE or Phos-tag SDS-PAGE analysis, respectively. (**a**) A549 cells were treated with EGF or OA, or transfected with PKM2 siRNA or HA-PKM2-expressing plasmid. (**b**) A549 cells were treated with pTyr, FBP or serine. Data are presented as the mean ± s.e.m. of three independent experiments. *$P < 0.05$. **$P < 0.01$ as determined by the one-way ANOVA test.

To confirm the association of PKM2 and SNAP-23 during the exosome release process, we performed the cross-immunoprecipitation (IP) experiment using anti-SNAP-23 and anti-PKM2 antibodies. In this experiment, A549 cell lysate was subjected to pre-clearance with control IgG and then incubated with anti-SNAP-23 or anti-PKM2 antibodies for IP. The immunoprecipitated products were blotted with anti-PKM2 or anti-SNAP-23 antibodies in a cross manner. As shown in Fig. 4c, immunoprecipitating SNAP-23 by anti-SNAP-23 antibody pulled down not only SNAP-23 but also PKM2 (left panel). As a control, neither protein was detected in IgG-immunprecipitated complex. In a similar fashion, immunoprecipitating PKM2 by anti-PKM2 antibody pulled down not only PKM2 but also SNAP-23 (right panel). The double IF and cross-IP results collectively suggest that PKM2 is associated with SNAP-23 during tumour cell exosome release.

**Phosphorylation of SNAP-23 mediates tumour exocytosis.** Recent studies have shown that dimerized phosphorylated-PKM2 possesses low catalytic activity in converting PEP to pyruvate but can serve as a protein kinase in phosphorylating other molecules such as STAT3 (ref. 21), histone H3 (refs 32,34) and MLC2 (ref. 40). Given that SNAP-23 phosphorylation is required for exocytosis and PKM2, particularly phosphorylated PKM2, is associated with SNAP-23 during exosome secretion, we speculated that PKM2 might promote tumour cell exosome secretion through phosphorylating SNAP-23. To test this hypothesis, we assessed the level of phosphorylated SNAP-23 in

A549 cells after modulating cellular PKM2 level via EGF or OA treatment (fetal bovine serum (FBS) as control), or transfection with plasmids expressing HA-PKM2 (CTL vector as control) or PKM2 siRNA (ncRNA as control). As shown in Fig. 5a, increase of PKM2 level via direct overexpression of PKM2 or EGF treatment significantly elevated the level of phosphorylated SNAP-23. In contrast, decrease of PKM2 level via transfection with PKM2 siRNA or OA treatment reduced the level of SNAP-23 phosphorylation. Moreover, direct alteration of dimeric or tetrameric formation of PKM2 obtained similar results. As shown in Fig. 5b, when A549 cells were treated with pTyr to increase the level of dimerized PKM2, the level of phosphorylated SNAP-23 was markedly elevated. In contrast, when treating A549 cells with FBP or serine to facilitate PKM2 tetrameric formation, the level of phosphorylated SNAP-23 was strongly reduced. These results indicate that phosphorylated PKM2, that is, dimerized PKM2, is involved in SNAP-23 phosphorylation.

We next sought to test whether dimerized PKM2 could directly phosphorylate SNAP-23. An *in vitro* phosphorylation assay was performed using both the recombinant SNAP-23 (rSNAP-23) and the recombinant PKM2 (rPKM2) purified from nuclear extracts of SW620 cells[21]. Since PKM2 uses PEP instead of ATP as a phosphate donor to phosphorylate ADP in the glycolysis, we replaced ATP by PEP in the *in vitro* reaction. After incubation under various conditions at room temperature for 1 h, the reaction mixtures were then subjected to SDS-PAGE or Phos-tag SDS-PAGE analysis detection of SNAP-23 phosphorylation. As shown in Fig. 6a, WB analysis demonstrated that the

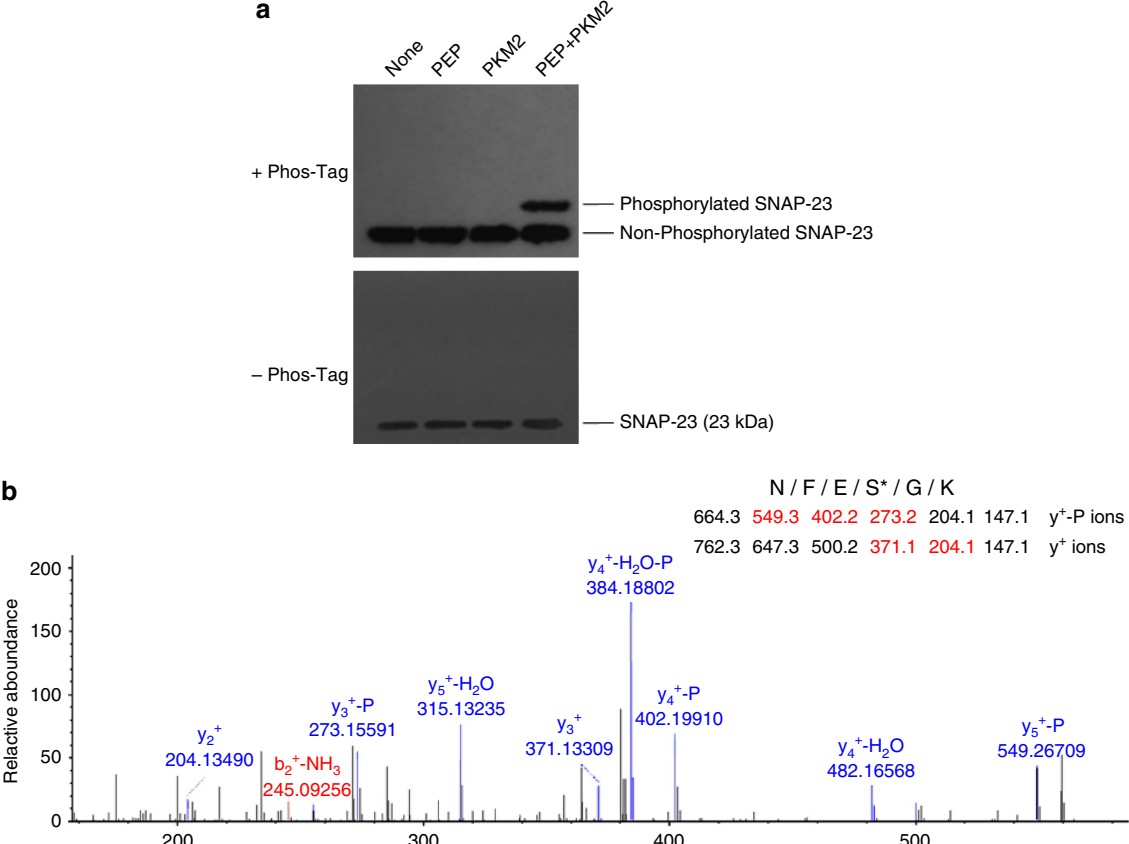

**Figure 6 | Direct phosphorylation of recombinant SNAP-23 (rSNAP-23) at Ser95 by recombinant PKM2 (rPKM2).** (**a**) Direct phosphorylation of rSNAP-23 by rPKM2. The rSNAP-23 was incubated with or without PEP, rPKM2 or PEP plus rPKM2 at room temperature for 1 h. The reaction mixtures were then subjected to SDS-PAGE or Phos-tag SDS-PAGE analysis. SNAP-23 was detected by anti-SNAP-23 antibody in WB analysis. (**b**) Phosphorylated SNAP-23 by rPKM2 analysed by mass spectrometry (MS). Note that MS analysis of tryptic fragment of rSNAP-23 treated with PEP/rPKM2 matches to the peptide [92]NFESGK[97] of SNAP-23, suggesting that SNAP-23 Ser95 was phosphorylated.

rSNAP-23 was phosphorylated by the rPKM2 in the presence of PEP, confirming that PKM2 acts as a protein kinase to remove the phosphate group from PEP and puts the phosphate on SNAP-23.

To identify the phosphorylation site on SNAP-23 used by PKM2, we further performed mass spectrometry (MS) analysis of purified recombinant SNAP-23 after phosphorylation assay (http://proteomecentral.proteomexchange.org, accession code: PXD005204). After fragmentation using trypsin, MS analysis identified a phosphorylated fragment matched to the peptide [92]NFESGK[97], suggesting that Ser95 was phosphorylated (Fig. 6b). The theoretical mass-to-charge ratio of ions with Ser95 phosphorylation ($Y^+$ ions) and Ser95 dephosphorylation ($Y^+$-P ions) are listed in Fig. 6b. There were five ions detected and marked in red.

To further examine the role of phosphorylation of SNAP-23 by PKM2 in mediating tumour cell exosome release, we constructed three plasmids expressing SNAP-23 mutants. The Ser95 of wild-type (WT) SNAP-23 was replaced with Glu95 (SNAP-23 (Ser95→Glu95)), whose carbolyic acid side chain will mimic the effect of phosphorylation. In contrast, to render a constitutively dephosphorylated state, we replaced Ser95 of WT SNAP-23 with Ala95 (SNAP-23 (Ser95→Ala95)). To ensure that serine phosphorylation by PKM2 is the critical factor (as opposed to phosphorylation of some other residue) enabling the role of SNAP-23 in exosome exocytosis, we also mutated Ser20 of SNAP-23 to Glu20 (SNAP-23 (Ser20→Glu20)). In addition to generating three mutated versions of SNAP-23 DNA, we also

generated siRNA-resistant constructs for each of our three mutated SNAP-23 plasmids. As shown in Figs 3 and 7a nucleotides within the binding sequence of SNAP-23 siRNA on SNAP-23 transcript were mutated to prevent siRNA binding without changing the amino acid sequence. As these His-tagged SNAP-23-expressing constructs are resistant to the effect of SNAP-23 siRNA, we designed them as R-SNAP-23 and R-SNAP-23 (Ser95→Ala95), respectively. WT SNAP-23 and SNAP-23 mutants were then expressed into the A549 cells and the release of exosomes at 24 h post-incubation was assayed by NTA. We found that knockdown of cellular SNAP-23 level via SNAP-23 siRNA significantly decreased exosome secretion (Fig. 7b). However, transfecting cells with R-SNAP-23 plasmid completely recovered the exosome secretion level. In contrast, transfecting cells with R-SNAP-23 (Ser95→Ala95) plasmid, which express an SNAP-23 protein that cannot be phosphorylated, failed to recover exosome secretion. Taken together, these results suggest that lack of phosphorylation of Ser95 on SNAP-23 would impair exosome secretion. The role of SNAP-23 Ser95 phosphorylation in promoting tumour cell exosome secretion was confirmed in cells transfected with the His-tagged SNAP-23 (Ser95→Glu95) construct that mimics constitutively phosphorylated SNAP-23 at Ser95. As shown in Fig. 7c, knockdown of cellular PKM2 via PKM2 siRNA markedly decreased exosome secretion. However, the reduction of exosome secretion by PKM2 siRNA was completely recovered by transfecting cells with His-tagged SNAP-23 (Ser95→Glu95) plasmid, which express an SNAP-23 form that is in constitutively phosphorylated state.

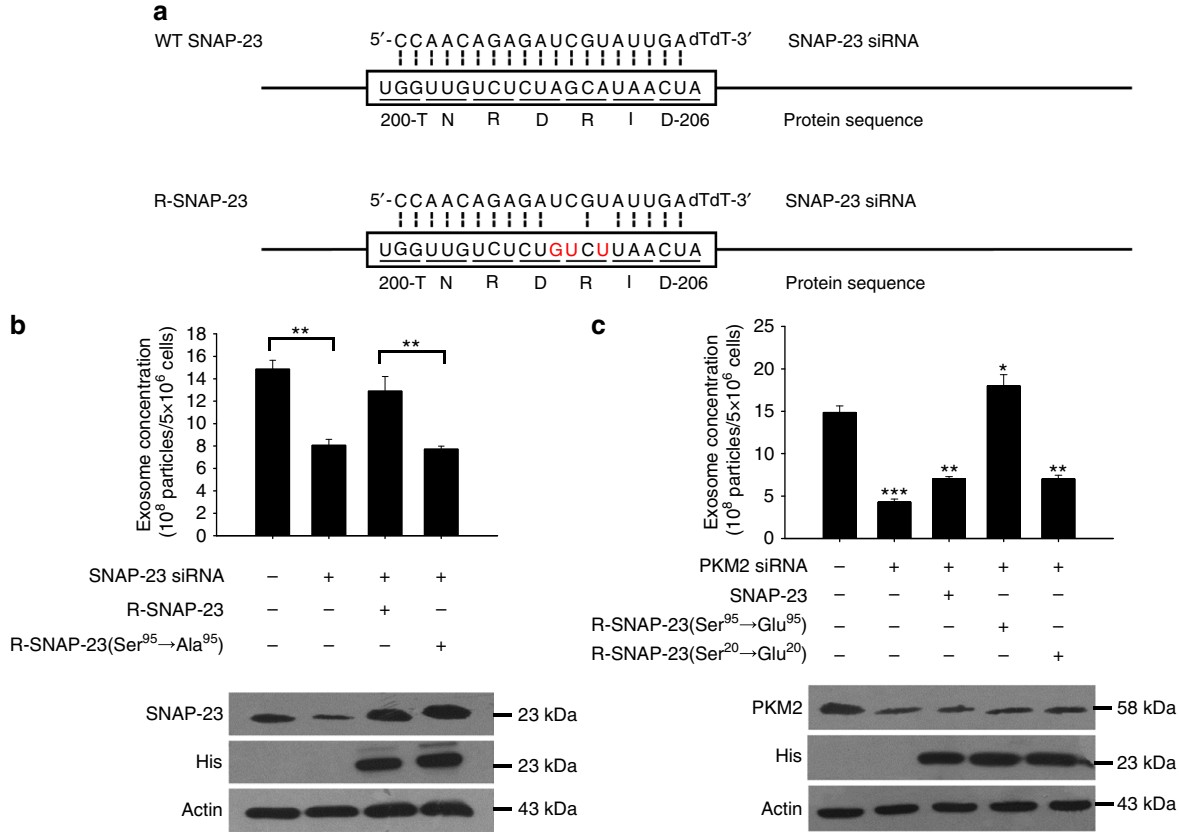

**Figure 7 | Phosphorylation of SNAP-23 at Ser95 by PKM2 promotes tumour cell exosome release.** (**a**) Constructs of His-tagged RNAi-resistant SNAP-23 (R-SNAP-23) and SNAP-23 (Ser95→Ala95) (R-SNAP-23 (Ser95→Ala95)), in which three nucleotides in siRNA target sequence were mutated without altering amino acid sequence. (**b,c**) Effect of SNAP-23 phosphorylation at Ser95 on PKM2-mediated release of exosomes from A549 cells. A549 cells were transfected with PKM2 siRNA and His-tagged plasmids expressing SNAP-23, R-SNAP-23, R-SNAP-23 (Ser95→Ala95), SNAP-23 (Ser20→Glu20) or SNAP-23 (Ser95→Glu95). The cellular protein levels of SNAP-23, PKM2 and His tag and the exosome concentration in cell culture medium were assayed by WB and NTA after 24 h culture, respectively. Data are presented as the mean ± s.e.m. of three independent experiments. $*P < 0.05$. $**P < 0.01$. $***P < 0.001$ as determined by the one-way ANOVA test.

To test whether serine phosphorylation other than Ser95 in SNAP-23 also plays a role in enhancing exosome secretion, we assessed the release of exosomes from cells co-transfected with PKM2 siRNA and His-tagged SNAP-23 (Ser20→Glu20) plasmid, in which Ser20 was mutated to Glu20 to force SNAP-23 in a constitutively phosphorylated form. As shown, SNAP-23 (Ser20→Glu20) did not recover the reduction of exosome secretion observed when PKM2 was knocked down with siRNA, further attesting to the serine phosphorylation at residue Ser95 of SNAP-23 serving as the critical factor in allowing exosome secretion.

## Discussion

It has been well established that tumour cells have elevated rates of glucose uptake and high lactate production in the presence of oxygen, known as aerobic glycolysis (also termed as the Warburg effect)[41]. Switch to glycolysis facilitates the uptake and incorporation of nutrients into nucleotides, amino acids and lipids to meet the requirements of rapid proliferation by tumour cells[42]. High lactate production also remodels the tumour microenvironment by contributing to angiogenesis, acting as a cancer cell metabolic fuel and inducing immunosuppression[43–45]. Apart from Warburg effect, an increased secretion of EVs, which can shuttle reciprocal signals and other molecules between transformed and stromal cells, is another phenomenon observed during tumorigenesis[6,46,47]. However, few studies have explored the link between the enhanced aerobic glycolysis and active exosome secretion in tumour cells.

In the present study, we have shown that PKM2 is the key factor modulating both glycolysis and exosome release. PKM2, an enzyme whose expression is upregulated in tumour cells whose metabolic needs are met through glycolysis, promotes the release of exosomes from tumour cells by phosphorylating Ser95 of SNAP-23, a component of the synaptosome/SNARE complex. First, we demonstrated a positive link between the cell's reliance on glycolysis and exocytosis of exosomes. Moreover, when we inhibited glycolysis, then the exosome release was attenuated. Second, we showed, utilizing knockdown or overexpression of PKM2 in a cell, that PKM2 is pivotal for exosome release. Using various molecules to promote dimerization or tetramerization of PKM2, we showed that dimerization is relevant to exosome release from the cell. Third, we showed that PKM2 promotion of exosome release requires the presence of SNAP-23. Finally, we identified Ser95 as the critical residue in SNAP-23 that is phosphorylated by PKM2 in effecting exosome exocytosis. Collectively, these results confirm that PKM2 phosphorylation of SNAP-23 at Ser95 is a key to SNARE complex formation and the secretion of exosomes in tumour cells.

The biogenesis and release of exosomes is a multi-step process modulated by various molecules. Previous studies have shown that sphingolipid ceramide, nSMase2 and Rab small GTPases, such as Rab27a and Rab27b, are involved in regulating different steps of the exosome secretion[8–10]. Interestingly, after knocking

down PKM2 in A549 cells, we failed to find significant alteration in the levels of Rab27a, Rab27b and nSMase2 (Supplementary Fig. 7), suggesting that PKM2's enhancement of exosome secretion in tumour cells is not mediated through changes in the expression of Rab27a, Rab27b and nSMase2. Our results identify the mechanism through which PKM2 enhances the release of exosomes: PKM2 phosphorylates SNAP-23 at Ser95 enabling the formation of the SNAP complex required for the docking and fusion between endosome-containing MVEs and plasma membranes.

PKM2 is best known for catalysing the last step in the process of glycolysis, the transfer of a phosphate group from PEP to ADP yielding the products of pyruvate and ATP. PKM2 exhibits a high level of catalysis for this reaction when protein is in a tetrameric state but not in a dimeric state. PKM2 is particularly likely to form a dimer when tyrosine 105 (Y105) is phosphorylated[34]. In its dimeric state, Yang et al.[32] found that PKM2 phosphorylates nuclear histone H3 to promote gene transcription. Additional functions of PKM2 have also been recognized. PKM2 was found to activate transcription of MEK5 by phosphorylating STAT3 at Y105 (ref. 18) and to regulate cytokinesis of tumour cells via phosphorylating MLC2 (ref. 33). A previous study also showed that IgE-mediated mast cell degranulation requires PKM2, although mechanism remains unclear[35].

As catalysis of the last step of glycolysis generally occurs in the cytoplasm, PKM2 particularly PKM2 in its tetrameric form mainly locates in the cytoplasm. In order to phosphorylate various transcription factors, phosphorylated/dimerized PKM2 is required to be relocated from the cytoplasm to the nucleus. In the present study, we suggest that dimerized/phosphorylated PKM2 can also re-locate at the vesicular structures close to the cell plasma membranes where it associates with the component of SNARE complex. Following exosome release from the plasma membrane, PKM2 is found in the secreted exosome fraction. Interestingly, location of PKM2 in secreted exosomes was also detected by Buschow et al.[32] in B cells. Given that PKM2 and SNAP-23 are widely expressed in various cell types, the PKM2-SNAP-23 signal pathway may play a critical role in promoting exocytosis in other cell types as well.

Our data as well as the data of others offer clues as to the specific environments in the plasma membranes at which SNARE complexes are likely to form. In our study, we found that PKM2 and SNAP-23 co-localize at certain punctate structures close to or on the cell surface (Fig. 4b, arrows). These punctates might be lipid rafts or secretory vesicles very close to lipid rafts on the plasma membrane. Previous study by Suzuki and Verma[20] suggest that SNAP-23 is highly enriched in the lipid raft fraction during stimulated degranulation, and the localization of SNAP-23 in lipid rafts facilitates the phosphorylation of SNAP-23 at Ser95 and Ser120 by IKK 2, leading to SNARE complex formation and degranulation of mast cells. As we did not observe the phosphorylation of SNAP-23 at Ser120 by PKM2 in the direct phosphorylation assay, the phosphorylation site of SNAP-23 by PKM2 may be not identical for a putative IKK consensus sequences for phosphorylation. Collectively, these studies suggest that PKM2 particularly phosphorylated/dimerized PKM2 can phosphorylate SNAP-23, a key component of exocytosis machinery, to promote tumour cell exosome secretion.

In summary, the findings presented here demonstrate a role for PKM2 in the release of exosomes from tumour cells. PKM2 phosphorylates a component of the SNARE complex, SNAP-23 at Ser95. This event is critical to the exocytosis of exosomes. Moreover, the release of exosomes from the tumour cell is highly correlated with the tumour cell's metabolic switch to aerobic glycolysis, a metabolic switch in which the perturbation of function of PKM2 is implicated.

## Methods

**Cells and reagents.** The human colorectal carcinoma SW480, cervical carcinoma Hela, lung carcinoma A549 and hepatocellular carcinoma HepG2 cell lines were purchased from the Institute of Biochemistry and Cell Biology, Shanghai Institutes for Biological Science, Chinese Academy of Sciences (Shanghai, China). Cells were maintained at 37 °C in a humidified 5% $CO_2$ incubator in Dulbecco's modified Eagle medium (DMEM) (Gibco, CA, USA) that contained 10% fetal bovine serum, 100 units $ml^{-1}$ of penicillin and 100 $\mu g\, ml^{-1}$ of streptomycin. Muscle and mammary gland tissues were collected from 12-week pregnant BALB/c mice (Nanjing University Animal Center (Nanjing, China). Six mice were killed for primary cell culture in each experiment. All animal maintenance and experimental procedures were carried out in accordance with the US National Institute of Health Guidelines for Use of Experimental Animals and approved by the Animal Care Committee of Nanjing University. The tissues were cut into pastes and digested with collagenase I–II/trypsin mixture at 37 °C, filtered through a 200 μm mesh filter and centrifuged at 300g for 5 min. Phos-tag acrylamide was purchased from NARD Institute Ltd (Amagasaki, Japan). OA, FBP and serine were purchased from Sigma Aldrich (St Louis, MO, USA). The phosphotyrosine peptide (GGAVDDDpYAQ FANGG) was synthesized and purchased from Chinese Peptide Company (Shanghai, China). Rabbit polyclonal antibodies that recognize Tsg101 (ab133586, 1:1000 for WB), SNAP-23 (ab3340, 1:1000 for WB and 1:500 for immuno-fluorescence), anti-phospho Y105 PKM2 (ab156856, 1:1000 for WB) and anti-PKM2 (ab38237, 1:1000 for WB) were purchased from Abcam (Hangzhou, China). A mouse monoclonal antibody against PKM2 (sc-365684, 1:100 for IP and 1:500 for immunofluorescence) was purchased from Santa Cruz (Shanghai, China). Polyclonal rabbit antibodies that recognize anti-CD9 (sc-9148, 1:1000 for WB) and anti-CD63 (sc-15363, 1:1000 for WB) were purchased from Santa Cruz. A rabbit monoclonal antibody against PKM1 (15821-1-AP, 1:1000 for WB) was purchased from Protein-Tech (Wuhan, China). Expression vector for PKM1 and recombinant PKM2 were gifts from Zhi-Ren Liu (Georgia State University, Atlanta, USA). HA-PKM2 plasmid was purchased from GeneCopoeia (Guangzhou, China). Specific siRNAs against PKM1 (5′-GCGUGGAGGCUUCUUAUAA-3′), PKM2 (5′-CCAUAAUCGUCCUCACCAA-3′) and SNAP-23 (5′-CCAACAGAGAUCGU AUUGA-3′) were purchased from RIBOBIO (Guangzhou, China). Recombinant SNAP-23 and expression vectors of wide type/mutant SNAP-23 were synthesized and purchased from Genscript (Nanjing, China).

**Isolation of exosomes.** Exosomes were collected from equivalent amounts of culture medium, conditioned by equivalent amounts of cells. When reaching 80% of confluency, the cell layers were rinsed with DMEM and refreshed with DMEM containing 2% exosome-depleted FBS. Both sequential centrifugation and exosome isolation kit methods were used to isolate exosome and for further analysis. For sequential centrifugation method, the medium was harvested 24 h after cell culture or transfection and exosomes were isolated from the media by three sequential centrifugation steps at 4 °C: 15 min at 500g to remove cells; 30 min at 10,000g to remove cell debris; and ultracentrifugation at 110,000g for 70 min (Beckman Ti70) to pellet exosomes. The pellet was re-suspended in phosphate-buffered saline (PBS) and centrifuged at 110,000g for 70 min to remove soluble serum and secreted proteins. For isolation kit method, the medium was harvested 24 h after cell culture or transfection and then centrifuged at 2,000g for 30 min to remove cells and debris. Supernatant containing the cell-free culture media was transferred to a new tube and added 0.5 volumes of the Total Exosome Isolation reagent (Invitrogen; 4478359). The culture media/reagent mixture was mixed well by vortex and the samples were incubated at 4 °C overnight. After incubation, samples were centrifuged at 10,000g for 1 h at 4 °C. Exosomes were contained in the pellet at the bottom of the tube and then re-suspended in PBS for various assays.

**Transmission electron microscopy assay.** For the transmission electron microscopy assay, the exosome samples were prepared as above described. Briefly, the exosome pellet was placed in a droplet of 2.5% glutaraldehyde in PBS buffer and fixed overnight at 4 °C. The exosome samples were rinsed three times in PBS for 10 min each and then fixed in 1% osmium tetroxide for 60 min at room temperature. Then, the samples were dehydrated through a graded series of ethanol concentration eventually reaching 100%. They were infiltrated with Epon resin (Ted Pella) in a 1:1 solution of Epon: propylene oxide overnight on a rocker at room temperature. The next day they were placed in fresh Epon for several hours and then embedded in Epon overnight at 60 °C. Thin sections were cut on a Leica EM UC7 ultramicrotome, collected on formvar-coated grids, stained with uranyl acetate and lead citrate and examined in a JEOL JEM 1011 transmission electron microscope at 80 kV.

**Nanoparticle tracking analysis.** The number and size of exosomes were directly tracked using the Nanosight NS 300 system (NanoSight Technology, Malvern, UK)[48], configured with a 488 nm laser and a high-sensitivity sCMOS camera. Exosomes re-suspended in PBS at a concentration of 5 μg of protein per ml were further diluted 100- to 500-fold to achieve between 20 and 100 objects per frame. Samples were manually injected into the sample chamber at ambient temperature. Each sample was measured in triplicate at camera setting 13 with acquisition time of 30 s and detection threshold setting of 7. At least 200 completed tracks were

analysed per video. The NTA analytical software version 2.3 was used for capturing and analysing the data.

**Phos-tag SDS-PAGE and western blot.** Phos-tag SDS-PAGE was performed with 7.5% polyacrylamine gels containing 50-100 μM Phos-tag acrylamine and 100-200 μM MnCl₂. After electrophoresis, Phos-tag acrylamine gels were washed with transfer buffer (50 mM Tris, 384 mM glycine, 0.1% SDS, 20% methanol) containing 1 mM EDTA for 10 min with gentle shaking as per the manufacturer's protocol. Proteins were transfected to PVDF membranes and probed with antibodies followed by horseradish peroxidase-conjugated secondary antibody. Immuno-detection was carried out with the Millipore Immobilon chemiluminescent horseradish peroxidase substrate. Normalization was conducted by blotting the same samples with an antibody against actin. All the uncropped data was shown in Supplementary Fig. 9.

**Cell-based assay for glucose uptake and lactate production.** The levels of glucose uptake by tumour or non-tumour cells were measured with a Glucose Uptake Cell-Based Assay Kit (Cayman Chemical, Ann Arbor, MI, USA). Cells were seeded in 96-well plates at a density of $1 \times 10^4$ cells per well. After 24 h, glucose uptake assays were performed according to the manufacturer's protocol. Relative fluorescence units were determined at 485–535 nm using a VARIOSKAN FLASH (Thermo). The levels of lactate production were examined with a Lactate Assay Kit (Biovision, Milpitas, CA, USA). Cells were plated in 100-mm culture dishes at a density of $1 \times 10^6$ cells per plate. After incubation for 24 h the culture medium was replaced with FBS-free DMEM. After further 8 h incubation, lactate assays were performed with culture media collected from each sample according to the manufacturer's protocol and the optical density was measured at 570 nm using a Multiskan EX (Thermo).

**Transfection of cells with plasmid and siRNA.** We designed three siRNA oligonucleotides for each target and tested the effects of three siRNA oligonucleotides on gene silence and exosome secretion prior to experiments. As shown in Supplementary Fig. 8, the siRNA oligonucleotides against one given target showed a similar inhibitory effect and the siRNA oligonucleotides with the best effect of gene silence for each target were selected in our experiments. The siRNA sequences are as follows: siPKM1, 5′-GCGUGGAGGCUUCUUAUAA-3′; siPKM2, 5′-CCA UAAUCGUCCUC ACCAA-3′; siSNAP-23, 5′-CCAACAGAGAUCGUAUUGA-3′. Cells were seeded in six-well plates or 10-mm dishes, and they were transfected the following day using Lipofectamine 2000 (Invitrogen; 11668) according to the manufacturer's instructions. Cells and exosomes were harvested 24 h after transfection for the following assays.

**Immunofluorescence.** Immunofluorescence microscopy was used to identify the subcellular localization of PKM2 and SNAP-23 in A549 cells. Cells were cultured on four-well chamber slides. At the time of harvest, cells were fixed with 4% paraformaldehyde and then permeabilized with 0.01% Triton X-100 for 10 min and subsequently probed with antibodies against PKM2 and SNAP-23 followed by incubation with fluorescent-tagged secondary antibodies (488 and 594 nm). All samples were treated with DAPI dye for nuclear staining (358 nm). For confocal microscopy, the Nikon C2 Plus confocal microscope was used.

**Co-IP.** Cells were lysed with lysis buffer (20 mM Tris-HCl, 150 mM NaCl, 0.5% Nonidet P-40, 2 mM EDTA, 0.5 mM DTT, 1 mM NaF, 1 mM PMSF and 1% Protease Inhibitor Cocktail from Sigma, pH 7.4) for 30 min on ice. The lysates were cleared by centrifugation (16,000g) for 10 min at 4 °C and then immunoprecipitated individually with anti-SNAP-23 antibody, anti-PKM2 antibody or normal IgG followed by protein G-Agarose beads. After the elution, the proteins were lysed in RIPA lysis buffer for WB assays.

**Chemical crosslinking reaction.** Cells were dissociated by Trypsin-EDTA solution, centrifuged at 1000 r.p.m. for 5 min and then washed by PBS and centrifuged as described above. The precipitation was re-suspended with 100 μl PBS gently and the concentration of total protein was determined by BCA assays. The suspension of cells was diluted to 2 mg ml⁻¹ with PBS, and a final volume of 500 μl was needed for the following chemical crosslinking assays. For crosslinking reaction with EDC/sulfo-NHS, 10 μl of a freshly prepared aqueous crosslinker solution was added into 500 μl cell suspension, with final concentrations of EDC and sulfo-NHS of 10 and 5 mM, respectively. The reaction mixtures were gently incubated at room temperature, and the reactions were quenched by the addition of 10 μl aqueous dithiothreitol (DTT), the final concentration of DTT was 20 mM. Finally, cells were centrifuged at 3000g and 4 °C for the following WB assays.

**Mass spectrometry analysis.** The protein sample was loaded and separated by SDS-PAGE. Upon accomplishment by electrophoresis, the entire gel was fixed and stained with Coomassie Brilliant Blue for 1 h. Then the gel was detained in 30% acetonitrile/100 mM ammonium bicarbonate until the background was clean. Then the supernatant was removed, and the gel particles were freeze-dried. The sample

was then re-suspended in 100 mM DTT and incubated for 30 min at 56 °C for reduction, followed by treatment of 200 mM IAA for 20 min in the dark for alkylation. The sample was washed with 100 mM ammonium bicarbonate and 100% acetonitrile, and then lyophilized. About 2.5–10 ng μl⁻¹ Trypsin solution was added into the sample and the sample was incubated for 20 h at 37 °C. The remained gel pieces were sonicated for 15 min in extraction buffer and then merged with digested solution. The resulted peptide solution was lyophilized again, re-suspended in 0.1% formic acid and filtered. The peptide solution was auto-sampled onto the reverse phase trap column (75 μm ID × 2 cm, 5 μm C18), then was flushed onto the Thermo scientific EASY column (75 μm ID × 100 mm, 5 μm C18) for separation at a flow rate of 250 nl min⁻¹. The column temperature was maintained at 200 °C and interfaced with the mass spectrometer. Mobile phases A and B were 0.1% formic acid and 0.1% formic acid in 88% acetonitrile, respectively. The total gradient was set as follows: from 0 to 50 min, 4% B increase to 50%; from 50 to 54 min, the column was washed with 50% B to 100% B; then the 100% B was maintained for another 6 min. A Q-Exactive mass spectrometer (Thermo Fisher Scientific, San Jose, CA, USA) was used for MS analysis. The instrument operated at positive ion mode: MS1 spectral were collected at a resolution of 70,000, with an automated gain control (AGC) target of 3,000,000, and a max injection time of 50 ms. The $m/z$ range for MS1 full scan is 300–1,800. Precursors were filtered by quadrupole using isolation window of 2 Th. 10 MS2 spectra were collected after each MS1 scan at a resolution of 17,500, with AGC target of 1,000,000, and a max injection time of 100 ms. Precursors were fragmented by at a normalized collision energy of 27%.

**In vitro protein kinase assay.** The kinase reactions were performed as described previously[49]. In brief, bacterially purified recombinant PKM2 (200 ng) was incubated with SNAP-23 (100 ng) in kinase buffer (50 mM Tris-HCl (pH 7.5), 100 mM KCl, 50 mM MgCl₂, 1 mM Na₃VO₄, 1 mM dithiothreitol, 5% glycerol, 0.5 mM PEP and 0.05 mM FBP) in 25 μl at 25 °C for 1 h. The reactions were terminated by adding sodium dodecyl sulfate-polyacrylamide gel electrophoresis (SDS-PAGE) loading buffer and heating to 100 °C. The reaction mixtures were then subjected to phos-tag SDS-PAGE analyses.

**Statistical analysis.** The GraphPad Prism 5 and SigmaPlot 11.0 packages were used. Data are presented as the means ± standard error of the mean (s.e.m.). Technical as well as biological triplicates of each experiment were performed. Comparison between two groups was performed by Student's $t$-test. Multiple-group comparisons were determined using one-way ANOVA. A $P$-value $<0.05$ was considered statistically significant. Pearson correlation coefficient ($r$ value) was calculated assuming linear relationship between variables.

**Data availability.** The mass spectrometry proteomics data have been deposited to the ProteomeXchange Consortium (http://proteomecentral.proteomexchange.org) via the PRIDE partner repository with the dataset identifier PXD005204. All other remaining data is available within the Article and Supplementary Files, or available from the authors upon request

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

## Acknowledgements

We thank Dr Zhi-Ren Liu (Department of Biology, Georgia State University, Atlanta, GA, USA) for providing PKM1 and PKM2 recombinant proteins and expression constructs. We also thank Dr Jill Leslie Littrell (Georgia State University, Atlanta, GA, USA) for critical reading and constructive discussion of the manuscript. This work was supported by grants from National Basic Research Program of China (973 Program) (2014CB542300).

## Author contributions

K.Z. and Y.W. designed the study. Y.W., D.W., F.J., Z.B., L.L., C.P. and M.L. performed the experiments and analysed data. D.Z., X.C. and L.S. contributed to the materials for the study. C.-Y.Z., G.H. and Y.L. interpreted data and contributed to the discussion. K.Z. and Y.W. wrote the manuscript.

## Additional information

**Competing financial interests:** The authors declare no competing financial interests.

