## [Peer Review File · Nature Communications]

Reviewer #1 (Remarks to the Author)

The manuscript by Wei Y, Li M and Bian Z et al, reported an important and novel mechanism on exosome secretion in cancer cells, by which pyruvate kinase type M2 (PKM2) phosphorylated a SNARE protein, SNAP-23. The authors firstly found the relationship between exosome secretion amounts and aerobic glycolysis and also then that PKM2 regulated SNAP23 phosphorylation at Ser 95. They clearly demonstrate the evidence of the novel molecular mechanisms on release of exosomes in cancer cells, by performing overexpression and knocking-down of SNAP-23 and PKM2, and also by the immunoprecipitation of them.

Since tumor-derived exosomes govern cancer microenvironment and promote tumorigenesis, the focus of the paper is well phrased. Also, overall, the experiment was well performed, and the manuscript was well organized. However, the authors' descriptions of exosome secretion mechanism by SNAP-23 and PKM2 are unlikely to be of interest to general readers in Nature Communications. The authors should include more data such as in vivo therapeutic models, for example, using chemical inhibitors of PKM2 or SNAP-23 or the data showing clinical relevance regarding authors' finding.

In my view, this manuscript is more appropriate for publication in molecular biology journals such as Journal of Cell Science and Molecular Cellular Biology, although it certainly does not mean anything negative regarding this work itself.

Major comments:

1. Several reports have been published exosome secretion mechanisms. Ostrowski M et al. showed that Rab27a and Rab27b were involved in the different steps of the exosome secretion (PMID: 19966785). Trajkovic K et al. and Kosaka N et al. have reported that exosome biosynthesis is regulated by nSMase2 (PMID: 18309083 and PMID: 20353945). The data need to be provided to compare these gene effects with authors' finding.
2. It seems that the authors mixed up the terminology of exosomes and microvesicles, sometimes. It is still controversial but, exosomes are normally not belonged to microvesicles. They are generally different in size and also presumably secretion mechanism. In this field, extracellular vesicles (EVs) includes exosomes, microvesicles and apoptotic bodies.
3. How is the expression level or phosphorylation level of SNAP-23 in cancer cells or tissues?
4. It is not clear how the authors featured SNAP-23. As the authors described in the introduction, there are many other SNARE family genes involved in tumor cell.

Minor points:

1. In figure 1a, the authors showed EM images of exosomes in A549. But the picture quality is insufficient to show as a figure. Please revise them.
2. According to Materials and methods, the authors used t-test for all statistical analyses. However, obviously some of results showed multiple comparisons. Please revise.
3. In figure 2 c and d, they are opposite. In the Text, Fig. 2c is for EGF and fig. 2d is for OA, but in Figure, they are not.

Reviewer #2 (Remarks to the Author)

General comments:

This study by Zen and colleagues demonstrates a role of pyruvate kinase type M2 (PKM2) in tumor cell exosome release. They show that PKM2 phosphorylates SNAP-23 at Ser95 and promote the release of exosomes. This study suggests a non-metabolic function of PKM2 in exosome secretion. While the finding is interesting, there are a number of conceptual and technical issues that limit the conclusiveness of this study.

Major points:

1. The work focuses on exosomes release but not general exocytosis. It is inappropriate for the authors to claim exocytosis throughout the text.
2. The terms "microvesicles", "microparticles", and "exosomes" are misused to introduce the importance of exocytosis. "Microvesicles" and "microparticles" are usually used to describe the shedding vesicles, which are formed by direct budding from the plasma membrane rather than exocytosis.
3. In the first paragraph of the "Results" section. How do the authors define "rapid release of exosomes"? The data in this paragraph are insufficient to draw a conclusion that "release of exosomes in tumor cells requires high level of aerobic glycolysis". At least, culture medium containing different concentrations of glucose and inhibitors of aerobic glycolysis such as 2-deoxyglucose (2-DG) should be additionally used.
4. LLC tumor cells and 293T cells are used for the iTRAQ assay shown in the Supplementary Table S1. However, A549 and Hela cells were used for the studies in the main text. The iTRAQ assay should also use A549 and Hela cells.
5. The differential centrifugation protocol, which is currently the most commonly accepted method for exosome purification, should be performed in addition to the commercial Exosome Isolation Kit. Meanwhile, according to the described method, exosomes are purified from serum-containing medium in this study. However, it is well-known that serum also contains a considerable amount of exosomes, which may interfere with the detected concentrations of exosomes derived from cell cultures.

6. There is no evidence to support the statement in the Abstract that "During exocytosis, phosphorylated PKM2 dimerizes and is recruited to the cell's membrane where it phosphorylates synaptosome-associated protein 23 (SNAP-23) and enables the formation of the SNARE complex allowing release of the exosomes". By Figure 2a and 2b, the authors demonstrated that the level of total PKM2 was positively correlated with the amount of exosome release in tumor cells, while the non-tumor mammalian cells, mouse primary myoblast and mammary epithelial cell (MEC), express lower level of PKM2 and also secrete much less exosomes. But the phosphorylation levels of PKM2 in different cell lines and their correlation with exosome secretion levels were not studied. Also, after either PKM2 knockdown or PKM2 overexpression, only the expression level of total PKM2 but not p-PKM2 was studied. Thus, a link between phosphorylated PKM2 to exosome secretion levels is not supported.

7. As shown in Figure 4a, both PKM2 and p-PKM2 were found in exosomes derived from HeLa and A549 cells, which expressed higher level of PKM2. How about the levels of PKM2 and p-PKM2 in the exosomes derived from non-tumor mammalian cells and the tumor cells expressing lower PKM2?

8. Because apoptotic cells may secrete more exosomes than healthy cells, the effects of used inhibitors and PKM2 knockdown on cell apoptosis should be detected and described. Alternatively, stable PKM2-knockdown cells using shRNA should be used for the studies involving exosome collection.

9. In Figure 1a, the TEM picture for exosome characterization needs to be improved. In addition, immunogold-labeling with antibodies against exosomal markers should be provided.

10. In Figure 5a, what does CTL stand for? Different controls should be used for different treatments. For example, negative control siRNA should be used in parallel to PKM2 siRNA; a control HA vector should be used in parallel to HA-PKM2, while vehicle control should be used in parallel to OA and EGF treatments.

Minor points:

1. Figure legends should be improved since they are not very informative. For example, in Figure 1f, 1g and Figure 2j, the concentrations and time periods of drug treatments should be included. In Figure 1b and Figure 4a, the quality of loaded exosomal proteins for immunoblotting should be mentioned.

2. In Figure 1b, the molecular weight should be better indicated. This is especially important considering that CD63 usually shows a smeared glycosylation pattern (35-55 kD), which is quite different from the expression pattern shown in this figure.

3. In Figure 2g, to confirm the successful transfection of HA-tagged PKM2, anti-HA antibody should be used for immunoblotting.

4. The information of the used antibody against p-PKM2 should be provided.

Reviewer #3 (Remarks to the Author)

This study presents a convincing case that the metabolic enzyme, pyruvate kinase M2 (PKM2), acts as protein kinase to phosphorylate SNAP2 and drive tumour cell-exocytosis. The data presented are largely clear and support the central hypothesis. Although identification of PKM2 as a protein kinase is not new, this study does add to the growing repertoire of substrates and biological functions for this very dynamic protein. The paper is also nicely written, logically laid out and easy to follow. One potential weakness of the study is that all the work focuses on cancer cell-lines in culture. Some idea of whether these processes take place in a real cancer setting would be strengthening. None-the-less, I feel the paper makes a useful and interesting contribution. I do have some specific questions about the clarity of some of the experimental data and I have also identified some text errors, which can easily be addressed.

1. Fig 1 and associate text: The correlation between exosomes and glucose/lactate might be plotted as for Fig 2b.

2. Fig.4b and text169-177: The co-localisation microscopy is not very clear from the figure and it is also not clear what the arrow heads are highlighting. Could this figure be improved, perhaps by the inclusion of expanded panels or higher resolution images? Proximity ligation assays or FRET might be a more reliable way of demonstrating co-localisation in cells.

3. Fig 1g and Fig S1 should either both be included in the main figure OR both be put in supplementary. Separating them makes following the data more difficult.

4. In a number of experiments, siRNAs are used to suppress protein expression. Given the known problems with off-target effects using siRNA, it is becoming standard practice to use more than 1 oligo against a given target to show the same phenotype.

5. Line 184: The authors state 'almost equal amount of SNAP-23'. It is not clear what this means as western blot bands are only quantitative in a relative sense.

6. Lines 198-199 and Fig 5a: The reduced levels of SNAP-23 phosphorylation are not very convincing on the blots presented. Levels look similar for control, PKM2 siRNA and OA lanes. The quantitation suggests otherwise but a clearer blot would help convince.

7. Fig 7 and p10. The effects of R-SNAP-23 over-expression on siRNA knockdown efficiency needs to be examined and presented. Very often, over-expression of siRNA resistant constructs can significantly decrease the efficiency of knockdown of endogenous counterparts (presumably due to a degree of sequence overlap). The 'rescue' can simply reflect reduced siRNA efficiency, rather than expression of the rescue construct. Can the

authors present blots for experiments c and d to clarify knockdown and over-expression? An alternative strategy to target UTR sequences for knockdown of endogenous proteins can also overcome this potential problem.

8. Fig 7a is dispensable. Detailing the point mutations made is sufficient.
9. Discussion: The authors might discuss further the broader cancer implications of the link between a switch to glycolysis and the increase in exocytosis driven stromal remodelling.

Text modifications:

1. Line 51 - P- Selectin is misspelt
2. Line 59 - membrane^s of budding vesicles
3. Line 71 - 'In specific...' should be changed to 'Specifically ...'
4. Line 73: 'low catalytic activity but high protein kinase activity' should be changed as kinase activity is also a 'catalytic' activity. Also, '...forms dimer structure..' needs to be changed.
5. Line 78: should read '...via direct phosphorylation of SNAP-23.'
6. Lines 170-173: Conformation of the PKM1/2 data should be presented with the profiling data from the previous paragraph.
7. Fig. 5. Phosphorylate should be changed to either Phosphorylated or Phospho next to the panels in a and b.
8. Line 244: I think the authors mean Fig 7d not 7c

Point-to-Point Response to Reviewers and Editor

Reviewer #1

Major comments:

1. Several reports have been published on exosome secretion mechanisms. Ostrowski M et al. showed that Rab27a and Rab27b were involved in the different steps of the exosome secretion (PMID: 19966785). Trajkovic K et al. and Kosaka N et al. have reported that exosome biosynthesis is regulated by nSMase2 (PMID: 18309083 and PMID: 20353945). The data need to be provided to compare these gene effects with authors' findings.

Answer: We greatly appreciate reviewer's comment. We have cited these important works in the introduction section and briefly discussed this in the discussion section of our revision (line 59-67; line 320-326, highlighted in bright yellow). We have also tested whether PKM2/SNAP-23 regulates tumor cell exosome release through modulating Rab27a/b and nSMase2, critical molecules that have previously been shown to be involved in the different steps of the exosome secretion. As shown in the Fig. 1 below, we detected the expression level of Rab27a, Rab27b and nSMase2 in A549 cells after knocking down PKM2 via PKM2 siRNA and found that PKM2 knockdown did not affect the levels of Rab27a, Rab27b and nSMase2 in A549 cells. This result suggests that the role of PKM2 in promoting tumor cell exosome secretion is not through modulating the expression of Rab27a, Rab27b and nSMase2. It is possible that the PKM2-SNAP-23 system is not involved in the biogenesis or biosynthesis of exosomes but may guide the final docking and fusion step between endosome-containing MVs and plasma membranes.

Fig. 1. Western blot analysis of Rab27a, Rab27b and nSMase2 levels in A549 cells after PKM2 knockdown.

2. It seems that the authors mixed up the terminology of exosomes and microvesicles, sometimes. It is still controversial but, exosomes are normally not belonged to microvesicles. They are generally different in size and also presumably secretion mechanism. In this field, extracellular vesicles (EVs) includes exosomes, microvesicles and apoptotic bodies.

Answer: We thank reviewer for clarifying this critical issue. Accordingly, in our revision, we have corrected all the wrong usage of 'microvesicles' terminology (the changed parts are highlighted). Also, we replaced extracellular microvesicles with 'extracellular vesicles (EVs)' (line 45-46, highlighted).

3. How is the expression level or phosphorylation level of SNAP-23 in cancer cells or tissues?

Answer: We detected the level of SNAP-23 and p-SNAP-23 in mouse primary myoblast, mammary epithelial cell (MEC), SW480, HeLa, A549 and HepG2 cells by phos-tag SDS-PAGE and western blot. As shown in new Fig. 3a of revision, SNAP-23 was detected in both non-tumor and tumor cells. However, tumor cells particularly cells with higher capacity to secrete exosomes have a higher level of p-SNAP-23 compared to non-tumor cells. We have incorporated this result into the revision (line 177-180, highlighted in bright

yellow).

4. It is not clear how the authors featured SNAP-23. As the authors described in the introduction, there are many other SNARE family genes involved in tumor cell.

Answer: Selecting SNAP-23 as the SNARE protein for further study is mainly based on our proteomics analysis (iTRAQ and mass spectrometry) of isolated exosomes. We are sorry for missing such information in our original manuscript. In fact, we have performed iTRAQ or mass spectrometry analysis of exosomes secreted from five different cell types. SNAP-23 is the only SNARE family protein detected in all exosome samples though other SNARE members such as VAMP3/7 are also detected in some exosome samples. As shown in new supplemental Table S1, MS data detected only SNAP-23 but no other SNARE family genes in exosomes secreted from A549 cells.

Minor points:

1. In figure 1a, the authors showed EM images of exosomes in A549. But the picture quality is insufficient to show as a figure. Please revise them.

Answer: We have performed new EM as well as immune-gold label experiment using exosomes isolated from A549 cells. As shown in new Fig. 1a, better quality EM images show a clear double membrane structure of exosome, and immune-gold labeling of endosomal marker CD63 by anti-CD63 antibody (lower right image of panel a) shows the location of CD63 on the exosomal membranes.

2. According to Materials and methods, the authors used t-test for all statistical analyses. However, obviously some of results showed multiple comparisons. Please revise.

Answer: We have revised the statistical analyses and amended the methods according to reviewer's suggestion (line 508-512, highlighted in bright yellow).

3. In figure 2c and d, they are opposite. In the Text, Fig. 2c is for EGF and fig. 2d is for OA, but in Figure, they are not.

Answer: We have revised the text accordingly.

Reviewer #2

Major points:

1. The work focuses on exosomes release but not general exocytosis. It is inappropriate for the authors to claim exocytosis throughout the text.

Answer: We greatly appreciate reviewer for clarifying this critical issue. Accordingly, we have replaced 'exocytosis' with 'exosome release' or 'exosome secretion' throughout the manuscript (the changed parts are highlighted in bright yellow).

2. The terms "microvesicles", "microparticles", and "exosomes" are misused to introduce the importance of exocytosis. "Microvesicles" and "microparticles" are usually used to describe the shedding vesicles, which are formed by direct budding from the plasma membrane rather than exocytosis.

Answer: Again, we thank reviewer for clarifying this. We have corrected the terminology mix-up of exosomes and microvesicles throughout the manuscript (line 45-46, etc., the changed parts are highlighted in bright yellow).

3. In the first paragraph of the "Results" section. How do the authors define "rapid release of exosomes"? The data in this paragraph are insufficient to draw a conclusion that "release of exosomes in tumor cells requires high level of aerobic glycolysis". At least, culture medium containing different concentrations of

glucose and inhibitors of aerobic glycolysis such as 2-deoxyglucose (2-DG) should be additionally used.

Answer: We agree with reviewer that more data are needed to support our conclusion that release of exosomes in tumor cells requires high level of aerobic glycolysis. According to reviewer's suggestion, we performed two additional experiments to show the dependence of exosome secretion on tumor cell aerobic glycolysis. As shown in new supplemental fig. S1, culture medium containing high glucose concentration promoted A549 cell exosome secretion (fig. S1a), while the inhibitor of glycolysis 2-DG (10 mM) markedly reduced A549 cell exosome release (fig. S1b).

4. LLC tumor cells and 293T cells are used for the iTRAQ assay shown in the Supplementary Table S1. However, A549 and Hela cells were used for the studies in the main text. The iTRAQ assay should also use A549 and Hela cells.

Answer: According to reviewer's comment, we performed additional mass spectrometry analysis of exosome samples isolated from A549 cell culture medium. As shown in the new supplemental Table S1 (line 195-200), PKM2, SNAP-23 and other endosomal markers (TSG101, CD63 and CD81) or small GTPases that maybe related to exosome biogenesis (Rab2a and Rab14) are detected in exosomes secreted from A549 cells. Interestingly, SNAP-23 is the only SANRE family member detected in A549 exosomes.

5. The differential centrifugation protocol, which is currently the most commonly accepted method for exosome purification, should be performed in addition to the commercial Exosome Isolation Kit. Meanwhile, according to the described method, exosomes are purified from serum-containing medium in this study. However, it is well-known that serum also contains a considerable amount of exosomes, which may interfere with the detected concentrations of exosomes derived from cell cultures.

Answer: First, several studies comparing the morphology, protein markers and RNA composition of exosomes isolated by ultracentrifugation and commercial Exosome Isolation Kit reveal that two protocols are both qualified to isolate exosomes (Zeringer *et al. World journal of methodology*, 2013; Marques-Garcia *et al Methods in molecular biology*, 2016; Schageman *et al. BioMed research int*, 2013). Nevertheless, we agree with reviewer that more convinced sequential (differential) centrifugation protocol should be performed in addition to the commercial Exosome Isolation Kit. Accordingly, we repeated most experiments (EM, immune-gold labeling, mass spectrometry, exosome secretion, etc.) using exosomes isolated by differential centrifugation protocol, and obtained results similar to those from exosomes isolated by Exosome Isolation Kit. As shown in new supplemental fig. S5 (line 187-191), we detected the effects of PKM2 knockdown or overexpression on exosome release using exosomes isolated by differential centrifugation protocol. The results clearly showed that exosome release from A549 cells and Hela cells was significantly decreased after knocking down PKM2 via PKM2 siRNA (fig. S5a), while exosome secretion from myoblast and MEC was markedly enhanced following the overexpression of PKM2 via transfection with HA-PKM2 expressing plasmid (fig. S5b).

Second, we thank reviewer for pointing out that serum contains a considerable amount of exosomes. We had realized this when we found high stability of microRNAs encapsulated in exosomes in human and animal serum samples (Chen *et al, Cell Research*, 2008). In the present study, exosomes in fetal bovine serum were removed by ultracentrifugation and exosome-depleted FBS was used. We have added this information to the method (line 291, highlighted in bright yellow).

6. There is no evidence to support the statement in the Abstract that "During exocytosis, phosphorylated PKM2 dimerizes and is recruited to the cell's membrane where it phosphorylates synaptosome-associated protein 23 (SNAP-23) and enables the formation of the SNARE complex allowing release of the exosomes". By Figure 2a and 2b, the authors demonstrated that the level of total PKM2 was positively correlated with the amount of exosome release in tumor cells, while the non-tumor mammalian cells, mouse primary myoblast and mammary epithelial cell (MEC), express lower level of PKM2 and also

secret much less exosomes. But the phosphorylation levels of PKM2 in different cell lines and their correlation with exosome secretion levels were not studied. Also, after either PKM2 knockdown or PKM2 overexpression, only the expression level of total PKM2 but not p-PKM2 was studied. Thus, a link between phosphorylated PKM2 to exosome secretion levels is not supported.

Answer: We understand reviewer's concern about our claim. Due to lack of evidence to fully support the statement, we have revised the statement in the Abstract of revision. The revised sentence reads as: 'During exosome secretion, phosphorylated PKM2 becomes a protein kinase to phosphorylate synaptosome-associated protein 23 (SNAP-23), which in turn, enables the formation of the SNARE complex allowing release of the exosomes' (line 33-36). In addition, according to reviewer's comment, we detected the phosphorylated PKM2 (p-PKM2) levels in mouse primary myoblast cells, mammary epithelial cells, SW480, HeLa, A549 and HepG2 cells. As shown in new Fig. 2c and 2d, phosphorylated PKM2 levels are positively correlated with the total exosome secretion in tumor or non-tumor cells. As shown in new supplemental fig. S2, we have also detected p-PKM2 level in A549 and HeLa cells after PKM2 knockdown or myoblast and MEC after PKM2 overexpression, and the results show that knockdown or overexpression of PKM2 decreased or increased the cellular level of p-PKM2, respectively.

7. As shown in Figure 4a, both PKM2 and p-PKM2 were found in exosomes derived from HeLa and A549 cells, which expressed higher level of PKM2. How about the levels of PKM2 and p-PKM2 in the exosomes derived from non-tumor mammalian cells and the tumor cells expressing lower PKM2?

Answer: According to reviewer's comment, we assayed the levels of PKM2 and p-PKM2 in exosomes from different cells by western blot analysis. As shown in new supplemental fig. S6, both PKM2 and p-PKM2 were detected in exosomes derived from tumor cells SW480 and A549 cells but not from mouse primary myoblast cells. Interestingly, the exosomes from SW480, a tumor cell line expressing lower PKM2 than A549 cells, also had less amount of PKM2 or p-PKM2 than the exosomes from A549 cells.

8. Because apoptotic cells may secrete more exosomes than healthy cells, the effects of used inhibitors and PKM2 knockdown on cell apoptosis should be detected and described. Alternatively, stable PKM2-knockdown cells using shRNA should be used for the studies involving exosome collection.

Answer: We thank reviewer for this insight. Accordingly, we detected cell apoptosis after glycolysis inhibitor shikonin treatment or PKM2 knockdown by PI or Annexin V labeling and flow cytometry. As shown in new supplemental fig. S3, the population of early ($7.30 \pm 0.65\%$) and late ($6.1 \pm 0.48\%$) apoptotic cells increased to $13.76 \pm 1.1\%$ and $9.18\% \pm 0.75$ after $1 \mu\text{M}$ shikonin treatment (24 h post-treatment). After PKM2 knockdown, the population of early and late apoptotic cells increased to $21.83 \pm 2.53\%$ and $13.58 \pm 1.62\%$. Given that apoptotic cells may secrete more exosomes than healthy cells, increase of apoptotic cell population should lead to more exosome secretion. However, as the total exosome release is significantly reduced by shikonin treatment or PKM2 knockdown, these results collectively demonstrate that aerobic glycolysis and PKM2 play an essential role in promoting tumor cell exosome release.

9. In Figure 1a, the TEM picture for exosome characterization needs to be improved. In addition, immunogold-labeling with antibodies against exosomal markers should be provided.

Answer: As shown in new Fig. 1a of our revision, we performed additional EM and immune-gold labeling experiment. New TEM pictures clearly show the double-membrane structure of exosome, and immune-gold labeling (lower right image in panel a) indicates the location of endosomal marker CD63 at exosome membrane.

10. In Figure 5a, what does CTL stand for? Different controls should be used for different treatments. For example, negative control siRNA should be used in parallel to PKM2 siRNA; a control HA vector should be used in parallel to HA-PKM2, while vehicle control should be used in parallel to OA and EGF treatments.

Answer: According to reviewer's comment, we have repeated experiments using proper controls as reviewer suggested. The data were shown in new Fig. 5a of our revision. We have also specified what different CTL stand for (line 239, highlighted). For instance, the CTL for PKM2 siRNA stands for ncRNA oligonucleotide.

Minor points:

1. Figure legends should be improved since they are not very informative. For example, in Figure 1f, 1g and Figure 2j, the concentrations and time periods of drug treatments should be included. In Figure 1b and Figure 4a, the quality of loaded exosomal proteins for immunoblotting should be mentioned.

Answer: We have revised the text accordingly.

2. In Figure 1b, the molecular weight should be better indicated. This is especially important considering that CD63 usually shows a smeared glycosylation pattern (35-55 kD), which is quite different from the expression pattern shown in this figure.

Answer: We indicated the molecular weight of proteins in new Fig. 1b. Indeed, CD63 shows a smeared pattern of glycosylated protein.

3. In Figure 2g, to confirm the successful transfection of HA-tagged PKM2, anti-HA antibody should be used for immunoblotting.

Answer: As shown in Fig. 2 below, we detected the HA by western blot to confirm the successful transfection of HA-tagged PKM2.

Fig. 2. WB detection of HA after PKM2 overexpression in myoblast and MEC cells.

4. The information of the used antibody against p-PKM2 should be provided.

Answer: We provided the information of antibody against p-PKM2 in the manuscript (line 378-381).

References

- Zeringer, E. *et al.* Methods for the extraction and RNA profiling of exosomes. *World journal of methodology* **3**, 11-18 (2013).
- Marques-Garcia, F. & Isidoro-Garcia, M. Protocols for Exosome Isolation and RNA Profiling. *Methods in molecular biology (Clifton, N.J.)* **1434**, 153-167 (2016).
- Schageman, J. *et al.* The complete exosome workflow solution: from isolation to characterization of RNA cargo. *BioMed research international* **2013**, 253957 (2013).
- Chen, X. *et al.* Characterization of microRNAs in serum: a novel class of biomarkers for diagnosis of cancer and other diseases. *Cell Res.* **18**(10):997-1006 (2008).

Reviewer #3

1. Fig 1 and associate text: The correlation between exosomes and glucose/lactate might be plotted as for Fig 2b.

Answer: According to reviewer's comment, we showed the correlation between exosome concentration and glucose uptake/lactate production in new Fig. 1f of our revision.

2. Fig.4b and text169-177: The co-localization microscopy is not very clear from the figure and it is also not clear what the arrow heads are highlighting. Could this figure be improved, perhaps by the inclusion of expanded panels or higher resolution images? Proximity ligation assays or FRET might be a more reliable way of demonstrating co-localization in cells.

Answer: We understand reviewer's concern on the data of co-localization of PKM2 and SNAP-23. We thus performed new double labeling and confocal microscopy analysis. As shown in new Fig. 4b, better images show the partial co-localization of PKM2 and SNAP-23 at small punctate structures (arrows) in A549 cell.

3. Fig 1g and Fig S1 should either both be included in the main figure OR both be put in supplementary. Separating them makes following the data more difficult.

Answer: We thank reviewer for this constructive suggestion, and Fig. S1 has been inserted into main figures as new Fig. 1i.

4. In a number of experiments, siRNAs are used to suppress protein expression. Given the known problems with off-target effects using siRNA, it is becoming standard practice to use more than 1 oligo against a given target to show the same phenotype.

Answer: Indeed, we synthesized three oligonucleotides for targeting each gene in the present study. The siRNA sequences are as follows: siPKM1-1, 5'-ACUCAUGAGUACCAUGCGGA-3'; siPKM1-2, 5'-GCGU GGAGGCUUCUUAUAA-3'; siPKM1-3, 5'-CUUGCCUGCUGUGUCGGAG-3'; siPKM2-1, 5'-GCUGUG GCUCUAGACACUAAA-3'; siPKM2-2, 5'-GUUCGGAGGUUGAUGAAAUC-3'; siPKM2-3, 5'-CCAUA AUCGUCCUCACCAA-3'; siSNAP-23-1, 5'-CCAACAGAGAUCGUAUUGA-3'; siSNAP-23-2, 5'-AACUA AUGAUGCCAGAGAA-3'. The effects of gene silence by three siRNA oligonucleotides have been tested prior to experiments. As shown in Fig. 3 below, we treated A549 cells with 3 siRNA oligonucleotides for each target protein. Western blot showed the knockdown effects of the siRNA oligonucleotides. NTA assays showed the same phenotype of 3 siRNA oligonucleotides against one given target. The siRNA oligonucleotides with the best effect of gene silence are used to knock down PKM1, PKM2

Fig. 3. WB detection of gene silencing effects of siRNA oligonucleotides against PKM1 (a), PKM2 (b) or SNAP23 (c) in A549 cells. NTA assays showed a similar phenotype of 3 siRNA oligonucleotides against one given target. *P < 0.05. **P < 0.01.

or SNAP-23.

5. Line 184: The authors state 'almost equal amount of SNAP-23'. It is not clear what this means as western blot bands are only quantitative in a relative sense.

Answer: We have deleted the "almost equal amount" in the text. Also, we performed additional cross-IP and cross-blot assays, and new WB data were shown in new Fig. 4c.

6. Lines 198-199 and Fig 5a: The reduced levels of SNAP-23 phosphorylation are not very convincing on the blots presented. Levels look similar for control, PKM2 siRNA and OA lanes. The quantitation suggests otherwise but a clearer blot would help convince.

Answer: Phosphorylation of PKM2 and SNAP-23 has been assayed by SDS-PAGE or Phos-tag SDS-PAGE analysis numerous times in the present study. A new Phos-tag SDS-PAGE/SDS-PAGE image was provided in new Fig. 5a. The result showed that SNAP-23 phosphorylation level was decreased not only by PKM2 knockdown but also by OA treatment. In contrast, SNAP-23 phosphorylation level was increased not only by PKM2 overexpression but also by EGF treatment.

7. Fig 7 and p10. The effects of R-SNAP-23 over-expression on siRNA knockdown efficiency needs to be examined and presented. Very often, over-expression of siRNA resistant constructs can significantly decrease the efficiency of knockdown of endogenous counterparts (presumably due to a degree of sequence overlap). The 'rescue' can simply reflect reduced siRNA efficiency, rather than expression of the rescue construct. Can the authors present blots for experiments c and d to clarify knockdown and over-expression? An alternative strategy to target UTR sequences for knockdown of endogenous proteins can also overcome this potential problem.

Answer: According to reviewer's comment, we detected the protein level following the knockdown or overexpression process. As shown in new Fig. 7b and 7c, western blot analysis confirmed the protein knockdown or overexpression of SNAP-23 or mutants.

8. Fig 7a is dispensable. Detailing the point mutations made is sufficient.

Answer: We have removed Fig. 7a from the revision according to reviewer's comment.

9. Discussion: The authors might discuss further the broader cancer implications of the link between a switch to glycolysis and the increase in exocytosis driven stromal remodeling.

Answer: We appreciate reviewer's constructive comment. Accordingly, we have briefly discussed the potential cancer implication of the link between a switch to glycolysis and the increase in exocytosis driven stromal remodeling in the revision (line 298-306, highlighted).

Text modifications:

1. Line 51 - P- Selectin is misspelt

2. Line 59 - membrane^s of budding vesicles

3. Line 71 - 'In specific...' should be changed to 'Specifically ...'

4. Line 73: 'low catalytic activity but high protein kinase activity' should be changed as kinase activity is also a 'catalytic' activity. Also, '...forms dimer structure...' needs to be changed.

5. Line 78: should read '...via direct phosphorylation of SNAP-23.'

6. Lines 170-173: Conformation of the PKM1/2 data should be presented with the profiling data from the previous paragraph.

7. Fig. 5. Phosphorylate should be changed to either Phosphorylated or Phospho next to the panels in a and b.

8. Line 244: I think the authors mean Fig 7d not 7c

Answer: We thank reviewer for these comments and we have fixed all these errors in the revision accordingly.

Reviewer #1 (Remarks to the Author)

The authors addressed most of my concerns with the following exceptions: Authors prepared supplemental table S1 to explain the reason why they focused on SNAP23 among the family members. In their rebuttal, they mentioned that iTRAQ or mass spectrometry analysis of exosomes were conducted with five different cell lines and found that only SNAP23 was detected in all of them. The data is striking and clearly indicates the relationship between SNAP23 and PKM2. Although current supplemental table S1 is simply made with only A549 cell result, the authors should summarize five different cell data for iTRAQ or mass spectrometry and show it more comprehensively. Also, they added supplemental table S1 in the paragraph of figure 4, however, it should be in the paragraph of figure 3 because they featured SNAP23 from figure 3.

Reviewer #2 (Remarks to the Author)

The authors have performed most of the suggested experiments and addressed my concerns. The writing can be further improved.

Reviewer #3 (Remarks to the Author)

Following my initial review and suggestions for improvement, the authors have conducted further experiments and modified the manuscript. The points I made have been sufficiently addressed. A couple of minor issue remain but otherwise I feel the revised manuscript is a useful and interesting piece of work, worthy of publication in Nature Comms.

1. In response to point 4 where I addressed the need for multiple distinct siRNAs to validate phenotypes are caused by target knockdown, the authors present data in the rebuttal (Fig 3). It is still not clear why this is not included as either supplementary data or at the very least referred to in the text. It is important.

2. There are some sentence and editorial improvements to the text that would help. I think this manuscript could do with careful copy editing to remove small textual errors.

Here I have pointed out a few examples.

Line 30: 'involved in remodelling tumor-stromal environment' should perhaps read 'the remodelling of the tumour-stomal environment'

Line 35: '...the SNARE complex allowing release' should perhaps read '...the SNARE complex to allow release...'

Line 44: 'large amount of extracellular vesicles' It is grammatically incorrect to use amount here - it needs to be either 'number' or 'quantity'. Amount cannot be used for countable plural nouns.

Line 211: WB and western blot both used. Consistency required.

Point-to-Point Response to Reviewers and Editor

Reviewer #1

- 1. The authors addressed most of my concerns with the following exceptions: Authors prepared supplemental table S1 to explain the reason why they focused on SNAP23 among the family members. In their rebuttal, they mentioned that iTRAQ or mass spectrometry analysis of exosomes were conducted with five different cell lines and found that only SNAP23 was detected in all of them. The data is striking and clearly indicates the relationship between SNAP23 and PKM2. Although current supplemental table S1 is simply made with only A549 cell result, the authors should summarize five different cell data for iTRAQ or mass spectrometry and show it more comprehensively. Also, they added supplemental table S1 in the paragraph of figure 4, however, it should be in the paragraph of figure 3 because they featured SNAP23 from figure 3.*

Answer: We greatly appreciate reviewer's comment on this. Accordingly, we summarized five different cell data for iTRAQ or MS and showed it in Supplementary Table 1. Also we moved Supplementary Table 1 to paragraph of Figure 3 in the revision (page 8).

Reviewer #2

- 1. The authors have performed most of the suggested experiments and addressed my concerns. The writing can be further improved.*

Answer: We thank reviewer to help us to improve our work. As for the writing, Professor Jill Littrell from Georgia State University (Atlanta, GA) has proof-read and re-edited the manuscript. The writing has been improved.

Reviewer #3

- 1. In response to point 4 where I addressed the need for multiple distinct siRNAs to validate phenotypes are caused by target knockdown, the authors present data in the rebuttal (Fig 3). It is still not clear why this is not included as either supplementary data or at the very least referred to in the text. It is important.*

Answer: According to reviewer's suggestion, we have added the data into supplementary data as new supplementary Fig. 8. We have also described the way to select siRNA oligonucleotide and added siRNA sequences in the Materials and Methods section of the revision (page 19).

- 2. There are some sentence and editorial improvements to the text that would help. I think this manuscript could do with careful copy editing to remove small textual errors.*

Here I have pointed out a few examples.

Line 30: 'involved in remodelling tumor-stromal environment' should perhaps read 'the remodelling of the tumour-stomal environment'

Line 35: '...the SNARE complex allowing release' should perhaps read '...the SNARE complex to allow release...'

Line 44: 'large amount of extracellular vesicles' It is grammatically incorrect to use amount here - it needs to be either 'number' or 'quantity'. Amount cannot be used for countable plural nouns.

Line 211: WB and western blot both used. Consistency required.

Answer: We thank reviewer for correcting these errors for us, and these errors have been fixed in the revision accordingly (the changed parts are highlighted in bright yellow). Also, Prof. Littrell from Georgia State University (Atlanta, GA) did a final proofreading and re-edition, and we believe the writing has been improved.